# Compositional Flows for 3D Molecule and Synthesis Pathway Co-design

Tony Shen [* 1]   Seonghwan Seo [* 2]   Ross Irwin [3 4]   Kieran Didi [5 6]   Simon Olsson [4]   Woo Youn Kim [2]   Martin Ester [1]

## Abstract

Many generative applications, such as synthesis-based 3D molecular design, involve constructing compositional objects with continuous features. Here, we introduce Compositional Generative Flows (CGFlow), a novel framework that extends flow matching to generate objects in compositional steps while modeling continuous states. Our key insight is that modeling compositional state transitions can be formulated as a straightforward extension of the flow matching interpolation process. We further build upon the theoretical foundations of generative flow networks (GFlowNets), enabling reward-guided sampling of compositional structures. We apply CGFlow to synthesizable drug design by jointly designing the molecule's synthetic pathway with its 3D binding pose. Our approach achieves state-of-the-art binding affinity and synthesizability on all 15 targets from the LIT-PCBA benchmark, and $4.2\times$ improvement in sampling efficiency compared to 2D synthesis-based baseline. To our best knowledge, our method is also the first to achieve state of-art-performance in both Vina Dock (-9.42) and AiZynth success rate (36.1%) on the CrossDocked2020 benchmark.

## 1. Introduction

Sampling objects through *compositional* steps while modeling *continuous* state is essential for a wide range of scientific applications (Jain et al., 2023a; Wang et al., 2023). One such important application is synthesizable target-based drug design, which aims to jointly generate molecules through a sequence of compositional reaction steps and predict their continuous 3D conformations relative to a protein target (Li et al., 2022). To this end, we propose a flow-based generative framework that jointly models the compositional structure and continuous state of objects.

Diffusion models (Sohl-Dickstein et al., 2015; Ho et al., 2020; Song et al., 2021) and flow matching models (Lipman et al., 2023) have achieved state-of-the-art performance in high-dimensional modeling tasks such as 3D molecule generation and protein structure design (Hoogeboom et al., 2022; Campbell et al., 2024; Schneuing et al., 2024a). However, standard diffusion and flow matching are restricted to modeling all the dimensions of the object at once (Chen et al., 2024). This results in an inability to model the compositional structure of objects through sequential construction steps. As a consequence, two main limitation arise: (1) the validity of compositional objects cannot be ensured, as invalid generative actions cannot be masked, and (2) the potential for efficient reward credit assignment in the compositional space is restricted (Bengio et al., 2021; Hansen et al., 2022; Yao et al., 2023). In drug design, where synthesizability is crucial for wet-lab validation, molecules can be naturally viewed as compositional objects constructed through sequential synthesis steps. Unfortunately, existing diffusion and flow matching models lack the ability to effectively model and respect the compositional nature of synthesis constraints when generating molecules.

*Sequential models* are a natural fit for generating composite objects. For instance, autoregressive models have been applied for 3D molecular design (Peng et al., 2023; Gebauer et al., 2020). However, current autoregressive models lack mechanisms to correct errors from earlier steps, causing slight errors in early position predictions to cascade (Jin et al., 2022). Generative flow networks (GFlowNets; Bengio et al., 2021) have recently shown success in sampling compositional structure for synthesis-based molecule design (Koziarski et al., 2024; Cretu et al., 2024; Seo et al., 2024), but remain limited to 2D molecules.

In this paper, we identify the key gap in standard *flow matching* and *sequential models* when generating compositional objects with continuous properties. To address this limitation, we introduce **Compositional Generative Flows (CGFlow)**, a novel generative framework that enables flow-

---

[*]Equal contribution  [1]School of Computing Science, Simon Fraser University, Canada [2]Department of Chemistry, KAIST, Republic of Korea [3]Molecular AI, Discovery Sciences, R&D, AstraZeneca [4]Department of Computer Science and Engineering, Chalmers University of Technology, Sweden [5]Department of Computer Science, University of Oxford, UK [6]NVIDIA. Correspondence to: Tony Shen <tsa87@sfu.ca>.

*Proceedings of the 42nd International Conference on Machine Learning*, Vancouver, Canada. PMLR 267, 2025. Copyright 2025 by the author(s).

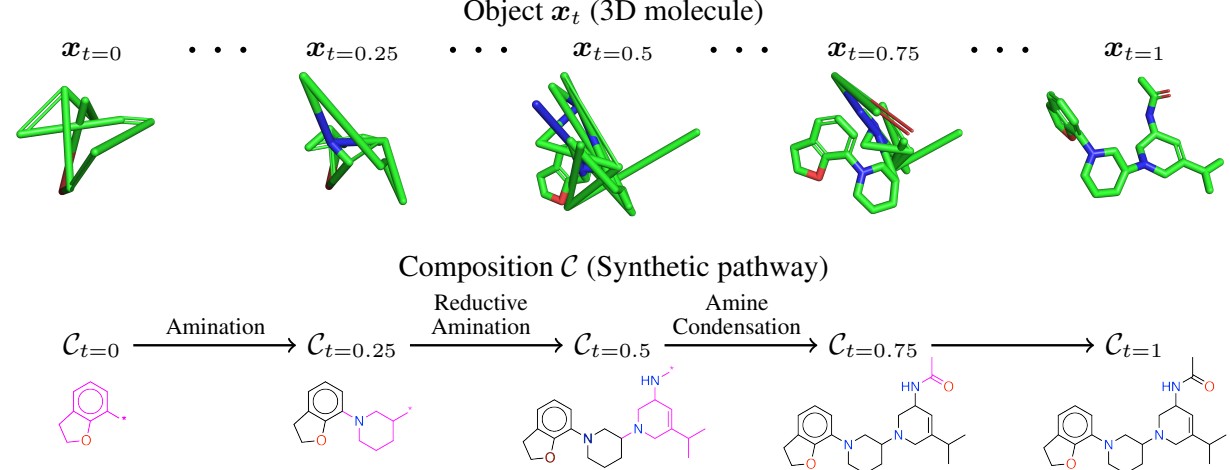

*Figure 1.* Overview of 3DSynthFlow generation. 3D Molecule $\boldsymbol{x} = (\mathcal{C}, \boldsymbol{S})$ consisting of its synthesis pathway $\mathcal{C}$ and 3D conformation $\boldsymbol{S}$ (visualized using $\boldsymbol{x}$). CGFlow generation interleaves 1). the continuous process for modeling position $\boldsymbol{S}$, and 2). the sequential sampling of synthesis steps at discrete intervals (t={0, 0.25, 0.5, 0.75}). The modeling of synthesis pathways and position are both dependent on the object $\boldsymbol{x}_t = \{\boldsymbol{S}_t, \mathcal{C}_t\}$, ensuring the interplay between the two processes.

based compositional generative modeling. Our key insight is that compositional generation in flow matching can be realized through a straightforward extension of the interpolation process to accommodate compositional state transitions. CGFlow represents a new generative modeling paradigm that combines the strengths of flow matching models for high-dimensional data while respecting the innate compositional nature of objects.

CGFlow consists of two interleaved flow processes: (1) *Compositional Flow* for modeling the probability path of compositional structure from data distribution to empty structures, and (2) *State Flow* for transporting the state variables associated with the compositional structure from data distribution to noise distribution.

To generate the interpolation path from data to noise, *Compositional Flow* progressively dismantles the compositional structure until it reaches an empty state. *State Flow* follows the standard optimal transport interpolation of flow matching, with the distinction that it assigns higher noise levels to the states of components removed earlier in the process, as described in Ruhe et al. (2024).

To generate compositional structures from the empty state, CGFlow samples constructive compositional steps with the generative policy, which can be modeled using distribution learning approaches, reinforcement learning, or GFlowNets. In particular, GFlowNets is used to prompt efficient exploration of compositional state spaces by reward-guided sampling (Bengio et al., 2021). The conditional flow matching (CFM) objective (Lipman et al., 2023) is used to estimate the vector field for generating state variables. Both compositional structure and state variables serve as inputs, ensuring

interdependence throughout the generative process.

As an application of the CGFlow framework, we present 3DSynthFlow, the method for target-based drug design ensuring synthesizability. We combine flow matching-based 3D structure generation with the GFlowNet-based synthesis-aware molecular generative model developed by Seo et al. (2024). Fig. 1 illustrates how 3DSynthFlow jointly generates the synthesis pathway (compositional structure) and 3D conformation (continuous state) of molecules. Previous flow-based generative models focused on generating either the 3D molecular structure or the synthesis pathway, but not both (see Sec. 5). 3DSynthFlow can jointly generate synthesis pathways and 3D molecular structures. This enables effective modeling of protein-ligand interactions and ensures synthesizability, both of which are essential for target-based drug discovery.

In our experiments, we evaluate 3DSynthFlow on the task of designing synthesizable drugs directly within protein pockets. 3DSynthFlow achieves 4.2× sampling efficiency improvement, and state-of-the-art performance across all 15 targets in the LIT-PCBA benchmark (Tran-Nguyen et al., 2020) for binding affinity. 3DSynthFlow can be extended to pocket-conditional setting and achieve state of-art-performance in both Vina Dock (-9.42) and AiZynth success rate (36.1%) on the CrossDocked benchmark.

Our contributions are summarized as follows:

- We propose Compositional Generative Flows (CGFlow), a flow-based framework that enables a generation of compositional objects while modeling continuous states.

- We incorporate GFlowNets in CGFlow to enable effi-

cient exploration of compositional state-space for high-rewarded samples.

- We use our compositional framework to develop 3DSynthFlow for 3D molecule and synthesis pathway co-design. 3DSynthFlow achieves $4.2\times$ improvement in sampling efficiency and demonstrate significant improvements in binding affinity and ligand efficiency on the LIT-PCBA benchmark.

- To the best of our knowledge, 3DSynthFlow is the first model to achieve state-of-the-art performance in both binding affinity and synthesis success rate on Cross-Docked2020.

## 2. GFlowNets preliminary

Generative Flow Networks (GFlowNets; Bengio et al., 2021) are a family of probabilistic models that learn a stochastic policy to construct compositional objects $x \in \mathcal{X}$ proportional to the reward of terminate state $R(x)$, i.e., $p(x) \propto R(x)$. Each object $x$ is constructed through a trajectory $\tau = (s_0 \to ... \to s_n = x) \in \mathcal{T}$ from the initial state $s_0$ and a series of state transitions $s \to s'$, where the terminate state is the object $s_n = x \in \mathcal{X}$.

A GFlowNet models a flow $F$ as an unnormalized density function along a directed acyclic graph (DAG) $\mathcal{G} = (\mathcal{S}, \mathcal{A})$, where $\mathcal{S}$ denotes the state space and $\mathcal{A}$ represents transitions. We define the *trajectory flow* $F(\tau)$ as a flow through the trajectory $\tau$. The *node flow* $F(s)$ is defined as the sum of trajectory flows through the node $s$, i.e., $F(s) = \sum_{s \in \tau} F(\tau)$, and the *edge flow* $F(s \to s')$ is defined as the total flow along the edge $s \to s'$, i.e., $F(s \to s') = \sum_{(s \to s') \in \tau} F(\tau)$.

From the flow network, we define two policy distributions. The *forward policy* $P_F(s'|s)$ executes the state transition $s \to s'$ from the flow distribution, i.e., $P_F(s'|s) = F(s \to s')/F(s)$. Similarly, the *backward policy* $P_B(s|s')$ distributes the node flow $F(s)$ to reverse transitions $s \dashrightarrow s'$, i.e., $P_B(s|s') = F(s' \to s)/F(s)$.

To match the likelihood of generating $x \in \mathcal{X}$ with the reward function $R$, two boundary conditions must be achieved. First, the node flow of each terminal state $x$, which represents the unnormalized probability of sampling the object $x$, must equal its reward, i.e., $F(x) = R(x)$. Second, the initial node flow $s_0$, which represents the *partition function $Z$*, must equal the sum of all rewards. i.e., $Z = \sum_{x \in \mathcal{X}} R(x)$. One such objective to satisfy these conditions is *trajectory balance* (TB; Malkin et al., 2023), defined as follows:

$$\mathcal{L}_{\text{TB}}(\tau) = \left( \log \frac{Z_\theta \prod_{t=1}^n P_F(s_t|s_{t-1}; \theta)}{R(x) \prod_{t=1}^n P_B(s_{t-1}|s_t; \theta)} \right)^2, \quad (1)$$

where the $P_F$, $P_B$, and $Z$ are directly parameterized to minimize the TB objective.

## 3. Compositional Generative Flows

CGFlow is a generative framework for modeling compositional objects with continuous states. The framework consists of two interleaved flows: the *Compositional Flow* for modeling compositional structures and the *State Flow* for modeling continuous states.

**Data representation.** Many applications naturally require compositional structure modeling, such as composing molecules via synthesis pathways (Gao et al., 2022) or constructing a causal graph (Nishikawa-Toomey et al., 2024).

Here, we represent an object $\boldsymbol{x}$ as a tuple $(\mathcal{C}, \mathcal{S})$, where $\mathcal{C}$ denotes the compositional structure and $\mathcal{S}$ represents the associated continuous states. For a given object, the compositional structure is defined as an ordered sequence $\mathcal{C} = (\mathbf{C}^{(i)})_{i=1}^n$, where $n$ denotes the number of its compositional components (e.g., molecular building blocks). The $i$-th component $\mathbf{C}^{(i)}$ is added at the $i$-th generative step of the trajectory $\tau$ (e.g., synthesis pathway).

Each compositional component $\mathbf{C}^{(i)}$ contains $m_i$ points (e.g., atoms) and $m_i$ may vary across components. Each point has an associated continuous state of dimension $d$ (e.g., atom position). $\boldsymbol{S}^{(i)}$ represents the states associated with component $i$ and is of size $(m_i, d)$. The continuous state $\boldsymbol{S}$ is defined as an ordered tuple of all states from each component $\mathcal{S} = (\boldsymbol{S}^{(i)})_{i=1}^n$.

This representation differs from standard flow matching, which models only state variables while ignoring the compositional structure and generation order of the object. In CGFlow, the compositional structure and state variables of an object are modeled jointly, ensuring the validity of the composition in generated objects.

### 3.1. Joint conditional flow process

We first define a joint conditional flow (Lipman et al., 2023) consisting of two components, the *compositional flow* and *state flow*. The joint conditional flow $\mathcal{P}_{t|1}(\cdot|\boldsymbol{x}_1)$ interpolates the object $\boldsymbol{x}$ from an initial state $\boldsymbol{x}_0$ to the final object $\boldsymbol{x}_1$. At the initial state, $\boldsymbol{x}_0 = (\mathcal{C}_0, \mathcal{S}_0)$, the compositional structure is represented as an empty graph, $\mathcal{C}_0 = \emptyset$, and the continuous state $\mathcal{S}_0 = [\,]$ has no dimension. The final object $\boldsymbol{x}_1 = (\mathcal{C}_1, \mathcal{S}_1)$ consists of its complete structure $\mathcal{C}_1$ and its continuous states $\mathcal{S}_1$. Our proposed flow must satisfy the following boundary conditions:

$$\mathcal{P}_{t|1}(\boldsymbol{x}_t|\boldsymbol{x}_1) = \begin{cases} \delta(\boldsymbol{x}_t = \boldsymbol{x}_0), & t = 0, \\ \delta(\boldsymbol{x}_t = \boldsymbol{x}_1), & t = 1. \end{cases} \quad (2)$$

which ensures that at $t = 0$, the flow starts at the initial state $\boldsymbol{x}_0$, and at $t = 1$, it reaches the final state $\boldsymbol{x}_1$.

### 3.1.1. COMPOSITIONAL FLOW

*Compositional Flow* defines a conditional probability flow over the compositional structure $\mathcal{C}$, progressively transitioning it from an empty graph $\mathcal{C}_0$ to a complete structure $\mathcal{C}_1$. Components are added sequentially in a predefined valid order, where $\mathbf{C}^{(1)}$ is the first component added. The function $k(t)$ determines how many components have been added at time $t$, defined as:

$$k(t) = \begin{cases} 0, & t = 0, \\ \min(\lfloor t/\lambda \rfloor + 1, n), & t > 0, \end{cases} \quad (3)$$

where $\lambda$ defines time interval between adding each compositional components. At time $t$, the compositional structure $\mathcal{C}_t$ includes the first $k(t)$ components from this order. $k(t)$ increases in discrete steps as $t$ progresses, starting from $k(0) = 0$ and ensuring $k(1) = n$, such that all $n$ components are added by $t = 1$. Each component $\mathbf{C}^{(i)}$ is generated at time $t_{\text{gen}}^{(i)} = \lambda \cdot (i - 1)$. To ensure that all components are generated within the valid time range, we require $\lambda \leq 1/n$ for all datapoints, satisfying $t_{\text{gen}} \leq 1 - \lambda$. The compositional structure at time $t$ is then given by:

$$\mathcal{C}_t = (\mathbf{C}^{(i)})_{i=1}^{k(t)}, \quad (4)$$

This formulation guarantees a gradual and sequential construction of $\mathcal{C}_t$, transitioning from the empty state $\mathcal{C}_0$ to the fully constructed structure $\mathcal{C}_1$ at fixed intervals.

Previous work models transitions across dimensions in diffusion processes using a rate function (Campbell et al., 2023). In contrast, we formulate transitions at fixed discrete intervals. Our approach retains key advantages of autoregressive generation, including simplified likelihood evaluation and learning objectives.

### 3.1.2. STATE FLOW

*State flow* defines a conditional probability path over the continuous states $\mathcal{S} = (\boldsymbol{S}^{(i)})_{i=1}^n$. Each continuous state $\boldsymbol{S}^{(i)}$ is initialized only when its corresponding compositional component $\mathbf{C}^{(i)}$ is generated, and components are generated at different times. Intuitively, we have greater uncertainty in the continuous states of recently added components than those generated earlier. Therefore, we introduce a temporal bias similar to that used in diffusion-based video generation (Ruhe et al., 2024): the global time $t$ is reparameterized into a component-wise local time $t_{\text{local}}^{(i)}$, defined as:

$$t_{\text{local}}^{(i)} = \text{clip}\left(\frac{t - t_{\text{gen}}^{(i)}}{t_{\text{window}}}\right), \quad (5)$$

where $t_{\text{gen}}^{(i)}$ is the generation time of component $\boldsymbol{C}^{(i)}$, $t_{\text{window}}$ is the interpolation time window, and $\text{clip}(x)$ ensures $t_{\text{local}}^{(i)} \in$

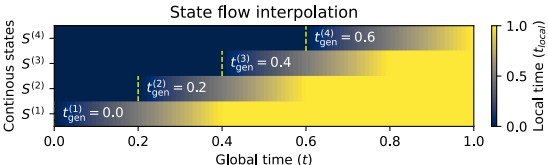

*Figure 2.* Local time for each component over time, with $n = 4$, $\lambda = 0.2$, and $t_{window} = 0.4$. $t_{local}^{(i)} = 0$ indicates the $S^{(i)}$ has not yet been initialized.

$[0, 1]$. Intuitively, $t_{\text{local}}^{(i)} = 0$ when the current time is $t = t_{\text{gen}}^{(i)}$, and $t_{\text{local}}^{(i)} = 1$ when the current time surpasses $t_{\text{gen}}^{(i)} + t_{\text{window}}$.

State flow is modeled as a linear interpolation based on $t_{\text{local}}^{(i)}$, combined with Gaussian noise applied throughout:

$$\mathbf{S}_t^{(i)} = \begin{cases} \mathcal{N}(t_{\text{local}}^{(i)} \mathbf{S}_1^{(i)} + (1 - t_{\text{local}}^{(i)}) \mathbf{S}_0^{(i)}, \sigma^2), & \text{if } t > t_{\text{gen}}^{(i)}, \\ [\,], & \text{else.} \end{cases}$$
$$(6)$$

where $\mathbf{S}_0^{(i)}$ represents the fully noisy initial state, and $\mathbf{S}_1^{(i)}$ is the final refined state. The continuous state $\mathbf{S}_t^{(i)}$ exists only if the corresponding component $\mathbf{C}^{(i)}$ has been generated, i.e., $t > t_{\text{gen}}^{(i)}$. To summarize, *state flow* enables the interpolation of continuous states when their associated compositional components are generated sequentially.

### 3.2. Sampling

During the sampling process, objects are generated by interleaving the integration of the *state flow* model $p_{1|t}^\theta$ and sampling actions using the *compositional flow* policy $\pi^\theta$. The process alternates between refining continuous states $\mathcal{S}_t$ and sequentially constructing the compositional structure $\mathcal{C}_t$ at fixed time points.

For state flow, the vector field governing the continuous states for the $i$-th component $\boldsymbol{S}^{(i)}$ is defined as $\hat{\boldsymbol{S}}_1^{(i)} - \boldsymbol{S}_t^{(i)}$, where $\hat{\boldsymbol{S}}_1^{(i)}$ is the predicted clean state by $p_{1|t}^\theta$, as formulated in previous works (Le et al., 2023). The rate at which we step in this vector field $\kappa^{(i)}$ is determined by the time remaining in the interpolation process for the state $\boldsymbol{S}^{(i)}$:

$$\kappa^{(i)} = \frac{\min(t_{\text{end}}^{(i)} - t, \Delta t)}{t_{\text{window}}}, \quad (7)$$

where $t_{\text{end}}^{(i)} = t_{\text{gen}}^{(i)} + t_{\text{window}}$ is the time at which the interpolation for component $i$ is completed, and $t_{\text{window}}$ is the interpolation window. The state values $\boldsymbol{S}^{(i)}$ are updated using Euler's method:

$$\boldsymbol{S}_{t+\Delta t}^{(i)} = \boldsymbol{S}_t^{(i)} + (\hat{\boldsymbol{S}}_1^{(i)} - \boldsymbol{S}_t^{(i)}) \cdot \kappa^{(i)} \Delta t, \quad (8)$$

Intuitively, if $t \geq t_{\text{end}}^{(i)}$, the state $\boldsymbol{S}_t^{(i)}$ is directly set to the predicted clean value $\hat{\boldsymbol{S}}_1^{(i)}$, ensuring the coherence of the continuous state.

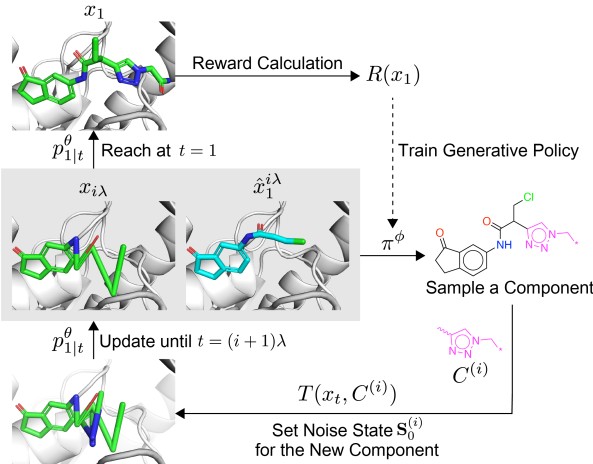

*Figure 3.* Overview of sampling and training using the pre-trained state flow model $p_{1|t}^\theta$. At $t = i\lambda$, compositional flow model $\pi^\phi$ samples a component $\boldsymbol{C}^{(i)}$ based on the state $x_t$ and its previous prediction $\hat{x}_1^t$. Then, the transition function $T$ incorporates the component $\boldsymbol{C}^{(i)}$ into the object. At $t = 1$, the compositional flow model is trained based on the reward of the generated object $x_1$.

For compositional structure generation, new components $\mathbf{C}^{(i)}$ are sampled from the compositional flow policy $\pi^\theta$ at discrete time intervals separated by $\lambda$. Transition function $T(\boldsymbol{x}_t, \mathbf{C}^{(i)})$ incorporates newly sampled component $\mathbf{C}^{(i)}$ into the object. $T$ also incorporates the new component's associated state $S_0^{(i)}$ by sampling its value from a noise distribution (typically Gaussian).

### 3.3. Training objectives

We train CGFlow to jointly approximate the distribution of continuous states $\mathcal{S}$ and the sequential generation of the compositional structure $\mathcal{C}$ with two models: (1) a state flow model $p_{1|t}^\theta$ for updating continuous states, and (2) a compositional flow policy $\pi^\theta$ for sampling compositional components. Both models are conditioned on the object $\boldsymbol{x}_t = (\mathcal{C}_t, \mathcal{S}_t)$ with self-conditioning $\hat{\boldsymbol{x}}_1^t = (\mathcal{C}_t, \hat{\mathcal{S}}_1^t)$ at time $t$, where $\hat{\mathcal{S}}_1^t = (\hat{\boldsymbol{S}}_1^{(i)})_{i=1}^{k(t)}$ from the previous prediction (Chen et al., 2022). This ensures their dependence on both the compositional structure and the continuous state of the object.

### 3.3.1. STATE FLOW LOSS.

The state flow model can be trained *simulation-free*, and independently of the compositional flow model. For each object $\boldsymbol{x}$, we first assign a generation trajectory $\tau$ over its compositional components $\mathcal{C}$. Then we can sample $\boldsymbol{x}_t$ according to the joint interpolation process (see Sec. 3.1).

The state flow model $p_{1|t}^\theta$ takes the entire object $\boldsymbol{x}_t = (\mathcal{C}_t, \mathcal{S}_t)$ as input and outputs $\hat{\mathcal{S}}_1^{t+\Delta t}$, the predicted clean continuous states. The objective sums over the continuous states $\boldsymbol{S}_t^{(i)}$ corresponding to components $\mathbf{C}^{(i)}$ that have

already been generated at the sampled time $t$:

$$\mathcal{L}_{\text{state}} = \mathbb{E}_{p_{\text{data}}(\boldsymbol{x}_1)\mathcal{U}(t)} \sum_{i=1}^{k(t)} \left\| p_{1|t}^\theta(\boldsymbol{x}_t)^{(i)} - \boldsymbol{S}_1^{(i)} \right\|_2^2, \quad (9)$$

where $k(t)$ represents the number of components generated at time $t$. The refinement model $p_{1|t}^\theta(\boldsymbol{x}_t)^{(i)}$ predicts $\hat{\boldsymbol{S}}_1^{(i)}$, the clean continuous states for component $i$, and we compute the MSE loss compared to the ground truth $\boldsymbol{S}_1^{(i)}$.

### 3.3.2. COMPOSITIONAL FLOW LOSS.

Given a pre-trained state flow model $p_{1|t}^\theta$, the compositional flow model $\pi_\phi$ samples a trajectory $\tau$ by sequentially adding a component $\boldsymbol{C}^{(i)}$ at $t = i\lambda$, conditioned on the current state $\boldsymbol{x}_t$ and self-conditioning $\hat{\boldsymbol{x}}_1^t$. This strategy allows the compositional flow model to make decisions using a less noisy estimate $\hat{\boldsymbol{x}}_1^t$, which is particularly advantageous in protein-ligand interaction modeling (Harris et al., 2023).

Inspired by GFlowNets (Bengio et al., 2021), we optimize the compositional flow model such that the probability of sampling a compositional structure for $\boldsymbol{x}_1$ is proportional to its reward $R(\boldsymbol{x}_1)$, using the trajectory balance (TB; Eq. (1)) objective. However, the object $\boldsymbol{x}_1 = (\mathcal{C}_1, \mathcal{S}_1)$ from a trajectory $\tau$ is stochastic due to the transition function $T$, which samples the initial state $\boldsymbol{S}_0^{(i)}$ from a noise distribution. To ensure a deterministic transition $T(\boldsymbol{x}_t, \boldsymbol{C}^{(i)})$, we fix the random seed when sampling the initial state $\boldsymbol{S}_0^{(i)}$.

Given a deterministic transition function and a fixed state flow model ODE, $\boldsymbol{x}_t = (\mathcal{C}_t, \mathcal{S}_t)$ is uniquely determined by a given trajectory $\tau$ sampled from the compositional flow model $\pi^\phi$. This formulation views the interpolation with the state flow model $p_{1|t}^\theta$ as the transition of the state resulting from the sampled action by the compositional flow policy $\pi^{\phi*}$. Then, TB loss is computed as:

$$\mathcal{L}_{\text{TB}}(\tau) = \left( \log \frac{Z_\phi \prod_{i=0}^{n-1} P_F(\boldsymbol{C}_{(i)} | \boldsymbol{x}_{i\lambda}, \hat{\boldsymbol{x}}_1^{i\lambda}; \phi)}{R(\boldsymbol{x}_1)} \right)^2. \quad (10)$$

We provide the theoretical background in Sec. B.

By enabling *online* training, this approach generalizes beyond the empirical data, as it can sample new data under the current policy. Alternatively, the compositional flow model can be trained using a maximum likelihood objective based on the data distribution (see Sec. A.3).

We refer readers to Sec. A.1 for a summary of the key steps in applying CGFlow to a new data domain. In the next section, we demonstrate CGFlow's application to 3D molecular generation and synthesis pathway design.

---

*Conceptually, it resembles an internal model of the world: used to predict the next state of the world as a function of the current state and action (Bryson & Ho, 1969). The state flow model weights are fixed to remain faithful to the data distribution.

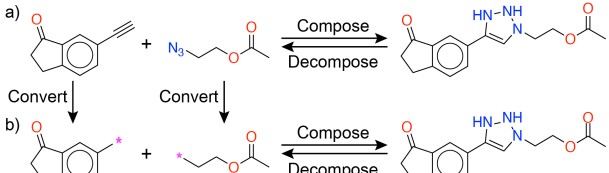

*Figure 4.* Illustration of synthesis-based composition rules. **(a)** reactant unit-based and **(b)** synthon unit-based (ours).

# 4. 3D Molecule and Synthesis Pathway Co-design within Protein Pockets

Synthesizability is a critical factor for ensuring molecules can be readily made for wet lab validation. Recent works have incorporated combinatorial chemistry principles into generative models to respond to this challenge (Koziarski et al., 2024; Cretu et al., 2024; Seo et al., 2024). Despite these advancements, these synthesis-based generative models are restricted to 2D molecular graphs, limiting their ability to capture the 3D protein-ligand interactions that are essential for biological efficacy.

To address this limitation, we introduce 3DSynthFlow, a generative method based on CGFlow, enabling the co-design of synthesis pathways and 3D molecular conformations. This approach facilitates effective modeling of protein-ligand interactions and synthesizability, which are both essential for target-based drug discovery.

**State flow model** predicts the docking pose of a molecule within a target protein pocket. Unlike typical docking, where the full 2D molecule is provided at $t = 0$, the state flow model must predict the pose for the ligand that is sequentially constructed at discrete time intervals. Molecules for training the state flow model $x$ are first decomposed into their compositional structure $\mathcal{C}$, which represents the sequentially composed synthesis pathway using Enamine reaction rules, and their continuous state $\mathcal{S}$, which represents the 3D coordinates refined over time. To ensure atoms not present in the molecule do not appear during decomposition (i.e., leaving group for reactions), we adopt a synthon-based representation rather than a reactant-based representation as Fig. 4. We discuss the motivation behind this formulation in Sec. E.1.2.

The state flow model is then trained using the objective described in Sec. 3.3.1 on the protein-ligand complex dataset, independent of the compositional flow model. The *Semla* architecture (Irwin et al., 2024), originally designed for 3D molecular generation, is used to parameterize the State flow model. We introduce new *hierarchical Semla* for memory-efficient protein encoding, and protein-ligand message passing to enable protein-ligand modeling (See Sec. D.2.1).

**Compositional flow model** generates a synthesis pathway to construct a molecular structure. The action space for

the composition flow model consists of all valid synthetic steps for a given structure $\mathcal{C}_t$ (see Sec. D.1). We modify the architecture from RxnFlow (Seo et al., 2024) to model the sampling policy for synthesis steps using 3D protein-ligand complexes as input (see Sec. D.2.2). The compositional flow model is trained online as described in Sec. 3.3.2.

# 5. Related Work

**Synthesis-based generative models.** Generative models have emerged as the key paradigm for discovering candidates by bypassing the expensive virtual screening. However, most generative models often render molecules outside the bounds of synthesizable chemical space, limiting their practical use in real-world applications (Gao & Coley, 2020). To address this limitation, several studies (Shen et al., 2024; Guo & Schwaller, 2024) have employed various synthetic complexity estimation methods (Ertl & Schuffenhauer, 2009; Coley et al., 2018; Kim et al., 2023; Neeser et al., 2024; Genheden et al., 2020) as the reward function.

Another promising direction is to design molecules by assembling purchasable building blocks under predefined synthesis protocols. (Gao et al., 2022; Li et al., 2022; Seo et al., 2023; Swanson et al., 2024; Gao et al., 2024). This strategy explicitly constrains the sample space to synthesizable chemical space. More recently, Koziarski et al. (2024); Cretu et al. (2024); Seo et al. (2024) have extended this strategy using GFlowNets, formulating synthesis pathway generation as trajectories of GFlowNets. This effectively explores the chemical space to discover diverse candidate molecules while balancing exploration and exploitation.

**Diffusion for sequential data.** Diffusion models have recently gained traction for generative modeling of sequential structures in diverse domains, spanning biological sequences (Campbell et al., 2024; Stark et al., 2024), videos (Ruhe et al., 2024), and language modeling (Lou et al., 2024). Campbell et al. (2023) propose a jump process for transitioning between different dimensional spaces to address the variable-dimension nature of data. To exploit the temporal causal dependency in sequences, Ruhe et al. (2024); Zhang et al. (2023) explore frame-level noise schedules for diffusion-based video generation for arbitrary-length frame rollout. Wu et al. (2023); Chen et al. (2024) apply similar ideas of training next-token prediction models while diffusing past ones for applications in planning and language modeling. Most similar to our work for molecule design using diffusion models are methods that use a separate diffusion process for each sequentially added fragment (Peng et al., 2023; Ghorbani et al., 2023; Li et al., 2024).

We provide an extended related works for sequential diffusion for molecular generation and structure-based drug design in Sec. C.

*Table 1.* **Average Vina docking score of Top-100 diverse modes.** We report two versions of SynFlowNet (v2024.05[a] and v2024.10[b]). Avg. and Med. are the average and median values over the average docking scores for all 15 LIT-PCBA protein targets. The results for the remaining 10 target proteins are reported in Appendix. 17 The best results are in bold.

| Category | Method | Average Vina Docking Score (kcal/mol, ↓) | | | | | | |
| --- | --- | --- | --- | --- | --- | --- | --- | --- |
| | | ADRB2 | ALDH1 | ESR_ago | ESR_antago | FEN1 | Avg. | Med. |
| Fragment | FragGFN | -10.19 (± 0.33) | -10.43 (± 0.29) | -9.81 (± 0.09) | -9.85 (± 0.13) | -7.67 (± 0.71) | -9.58 | -9.85 |
| | FragGFN+SA | -9.70 (± 0.61) | -9.83 (± 0.65) | -9.27 (± 0.95) | -10.06 (± 0.30) | -7.26 (± 0.10) | -9.22 | -9.58 |
| Reaction | SynNet | -8.03 (± 0.26) | -8.81 (± 0.21) | -8.88 (± 0.13) | -8.52 (± 0.16) | -6.36 (± 0.09) | -8.12 | -8.52 |
| | BBAR | -9.95 (± 0.04) | -10.06 (± 0.14) | -9.97 (± 0.03) | -9.92 (± 0.05) | -6.84 (± 0.07) | -9.35 | -9.84 |
| | SynFlowNet[a] | -10.85 (± 0.10) | -10.69 (± 0.09) | -10.44 (± 0.05) | -10.27 (± 0.04) | -7.47 (± 0.02) | -9.95 | -10.34 |
| | SynFlowNet[b] | -9.17 (± 0.68) | -9.37 (± 0.29) | -9.17 (± 0.12) | -9.05 (± 0.14) | -6.45 (± 0.13) | -8.78 | -9.17 |
| | RGFN | -9.84 (± 0.21) | -9.93 (± 0.11) | -9.99 (± 0.11) | -9.72 (± 0.14) | -6.92 (± 0.06) | -9.08 | -9.91 |
| | RxnFlow | -11.45 (± 0.05) | -11.26 (± 0.07) | -11.15 (± 0.02) | -10.77 (± 0.04) | -7.66 (± 0.02) | -10.46 | -10.84 |
| 3D Reaction | **3DSynthFlow** | **-11.96** (± 0.12) | **-11.82** (± 0.03) | **-11.58** (± 0.07) | **-11.23** (± 0.08) | **-7.79** (± 0.01) | **-10.89** | **-11.26** |

# 6. Experiments

**Overview** We evaluate 3DSynthFlow for synthesizable target-based drug design in two common settings: pocket-specific optimization (Sec. 6.1) and pocket-conditional generation (Sec. 6.2). Our experiments aims to address three main key questions: (1) Does 3DSynthFlow generate molecules with improved binding affinity and ligand efficiency compared to existing synthesis-based baselines? (2) Does co-designing 3D structures improve the sampling efficiency in discovering diverse high-reward modes? (3) How does 3DSynthFlow generalize to the pocket-conditional setting, and how does it compare to existing SBDD baselines?

To answer these questions, We first apply 3DSynthFlow to optimize for targets in the LIT-PCBA benchmark (Tran-Nguyen et al., 2020). We evaluate the affinity, ligand efficiency, synthesis success rate and protein-ligand interactions of generated molecules. Then, we compare the sampling efficiency of 3DSynthFlow against 2D-based baseline. Lastly, we comapre 3DSynthFlow against SBDD methods in the pocket-conditional setting on the CrossDocked dataset (Francoeur et al., 2020).

**Setup.** To construct synthetic pathway, we utilize 38 bimolecular Enamine synthesis protocols from Gao et al. (2024). We use Enamine's 1.2M-size catalog set and 300k-size stock set in Sec. 6.1 and Sec. 6.2, respectively. For stock set, we filtered out molecules with drug-likeness score lower than 60 (Lee et al., 2022). To ensure tractability, molecular generation is limited to two synthesis steps, consistent with Enamine REAL Space (Grygorenko et al., 2020). To estimate synthesizability, we employ the retrosynthetic analysis tool AiZynthFinder (Genheden et al., 2020).

We pre-train the state flow model for 3D pose prediction on the CrossDocked dataset (Francoeur et al., 2020) with LIT-PCBA targets removed (see Sec. E.3.2). We sample a synthesis pathway for each ligand using reaction rules to fragment the ligand (see Fig. 4). This exposes the state flow

model to intermediate states - allowing it to predict their poses for partial structures during compositional flow model training (see Sec. E.1.2 for motivation).

## 6.1. Pocket-specific optimization

**Setup.** We follow Seo et al. (2024) in both the reward function and evaluation protocol. To estimate the binding affinity, we use the GPU-accelerated docking tool UniDock (Yu et al., 2023). Each method generates up to 64,000 molecules for each protein target. To prevent *reward hacking* by increasing molecular size to improve the docking score, QED (Bickerton et al., 2012) is jointly optimized, and we set a generation of 40 heavy atoms. The reward function for multi-objective optimization is described in Sec. E.2.

Generated molecules are filtered based on property constraints (QED > 0.5), and the top 100 diverse modes are selected according to docking scores, ensuring structural diversity with a Tanimoto distance threshold of 0.5 [†]. Finally, we evaluated the average **Vina docking score** and **AiZynthFinder Success Rate** of the selected molecules. **Ligand effiency** as computed by (Vina / number of heavy atoms) is also reported to confirm the docking score improvement do not arise from simply increase in molecular size.

**Baselines.** We compare our approach against several synthesis-based methods, including a genetic algorithm **SynNet** (Gao et al., 2022), a conditional generative model **BBAR** (Seo et al., 2023), and multiple GFlowNets. We consider two settings of fragment-based GFlowNets, with and without synthetic accessibility score (SA; Ertl & Schufenhauer, 2009) objective (**FragGFN**, **FragGFN+SA**), and three different synthesis-based GFlowNets (**RGFN**, **SynFlowNet**, **RxnFlow**) (Koziarski et al., 2024; Cretu et al., 2024; Seo et al., 2024).

---

[†]Since we select top 100 modes filtering for similarity, diversity is not reported in this section.

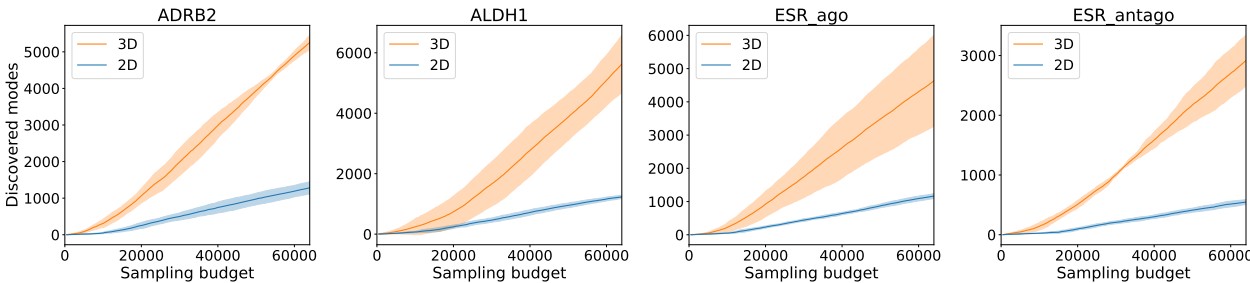

*Figure 5.* Number of discovered modes (satisfying Vina <-10 kcal/mol, QED >0.5, Sim <0.5) as a function of sampling budget for the first four LIT-PCBA targets for 3DSynthFlow (3D) vs RxnFlow (2D) across 3 seeds. Higher is better.

*Table 2.* The average success rate and synthetic steps estimated from AiZynthFinder for all 15 LIT-PCBA protein targets.

| Method | Success Rate (%, ↑) | Synthesis Steps (↓) |
|---|---|---|
| FragGFN+SA | 3.52 | 3.74 |
| SynNet | 47.50 | 3.45 |
| BBAR | 17.92 | 3.68 |
| SynFlowNet[a] | 54.60 | 2.55 |
| SynFlowNet[b] | 58.38 | 2.47 |
| RGFN | 47.43 | 2.46 |
| RxnFlow | 65.35 | **2.17** |
| **3DSynthFlow** | **68.58** | 2.39 |

*Table 3.* **PoseCheck** Averages and standard deviations over 4 runs, for the top 100 diverse modes for the first 5 LIT-PCBA pocket. The results for all protein-ligand interactions are reported in Appendix. F.5. The best results in each column are in **bold**.

| Method | H-Bond Acceptors | H-Bond Donors | Sum |
|---|---|---|---|
| SynFlowNet[a] | 0.22 (± 0.01) | 0.11 (± 0.01) | 0.33 |
| SynFlowNet[b] | 0.22 (± 0.03) | 0.10 (± 0.01) | 0.32 |
| RGFN | 0.19 (± 0.01) | 0.11 (± 0.01) | 0.30 |
| RxnFlow | 0.22 (± 0.00) | 0.10 (± 0.01) | 0.32 |
| **3DSynthFlow** | **0.33** (± 0.06) | **0.17** (± 0.03) | **0.50** |

**Results.** Table. 1 presents the Vina results for the first five targets, while the full property results for all targets are in Sec. F.7. 3DSynthFlow consistently outperforms all baselines across LIT-PCBA targets in affinity and ligand efficiency, demonstrating that co-designing 3D molecular structures alongside synthesis pathways enhances the discovery of high-affinity molecules. Table. 3 shows molecules generated by 3DSynthFlow also exhibit a higher number of protein-ligand interactions, as quantified by PoseCheck, highlighting the benefits of 3D-aware modeling for target-aware drug design (see full results in Sec. F.5).

Furthermore, Fig. 5 and Table. 16 presents the sampling efficiency for the first 5 LIT-PCBA targets. Diverse high-scoring modes were defined by QED>0.5, Vina<-10 kcal/-mol[‡], and Tanimoto similarity<0.5 to any other mode. After sampling 64,000 molecules, 3DSynthFlow identified 4.2x number of diverse high-scoring modes compared to Rxn-Flow (4432.3 vs 1062.6). This enhanced sampling efficiency validates the effectiveness of 3DSynthFlow framework, and suggests a higher probability of experimental success.

Table. 2 shows that 3DSynthFlow attains comparable synthesizability compared to its 2D baselines, demonstrating its generated molecules is likely to be synthesizable. This ensures that the high-affinity compounds identified by our model are not merely theoretical but represent viable candidates for experimental validation and further development.

Finally, we conduct extensive ablation studies on the various technical decisions regarding: flow matching steps (Sec. F.2), time scheduling (Sec. F.3), and use of pose-based rewards (Sec. F.1). Analysis on training efficiency can be found in Sec. E.1.3.

### 6.2. Pocket-conditional generation.

**Setup.** Our method generalizes to pocket-conditional generation problem setting (Peng et al., 2022; Guan et al., 2023; Schneuing et al., 2024a), enabling the design of binders for unseen pockets with a single model and no additional oracle calls. We follow the same pocket-conditional experimental setup and reward function used in TacoGFN (Shen et al., 2024) and RxnFlow (Seo et al., 2024) (see Sec. E.2. 3DSynthFlow adopt the pre-trained proxy from TacoGFN trained on CrossDock2020 train set, which leverages pharmacophore representation (Seo & Kim, 2024), to compute rewards for training a pocket-conditional policy.

Each method generates 100 molecules for each of the 100 test pockets in the CrossDocked2020 benchmark (Francoeur et al., 2020) and are evaluated on these additional metrics compared to the pocket-specific setting: **Diversity** represents the average pairwise Tanimoto distance computed from ECFP4 fingerprints (Morgan, 1965). We also report **Validity (%)**, the proportion of unique molecules that can be parsed by RDKit, and **Time (sec.)**, the average duration required to sample 100 molecules.

**Baselines.** We compare 3DSynthFlow with state-of-the-art distribution learning-based models trained on a syn-

---

[‡]Except for FEN1, where we use Vina below -7 kcal/mol to maintain similar number of modes compared to the other targets.

*Table 4.* **Benchmark Results for Generative Methods.** We report the average (Avg.) and median (Med.) values for each metric when available. Time indicates the average duration to generate 100 molecules. Reference denotes known actives. For methods where only one value is available, the median is indicated as "–".

| Category | Method | Validity (↑) | Vina (↓) | | QED (↑) | | AiZynth. Succ Rate (↑) | | Div (↑) | Time (↓) |
|---|---|---|---|---|---|---|---|---|---|---|
| | | Validity | Avg. | Med. | Avg. | Med. | Avg. | Med. | Avg. | Avg. |
| | Reference | - | -7.71 | -7.80 | 0.48 | 0.47 | 36.1% | - | - | - |
| Atom | Pocket2Mol | 98.3% | -7.60 | -7.16 | 0.57 | 0.58 | 29.1% | 22.0% | 0.83 | 2504 |
| | TargetDiff | 91.5% | -7.37 | -7.56 | 0.49 | 0.49 | 9.9% | 3.2% | 0.87 | 3428 |
| | DecompDiff | 66.0% | -8.35 | -8.25 | 0.37 | 0.35 | 0.9% | 0.0% | 0.84 | 6189 |
| | DiffSBDD | 76.0% | -6.95 | -7.10 | 0.47 | 0.48 | 2.9% | 2.0% | **0.88** | 135 |
| | MolCRAFT | 96.7% | -8.05 | -8.05 | 0.50 | 0.50 | 16.5% | 9.1% | 0.84 | 141 |
| | MolCRAFT-large | 70.8% | -9.25 | -9.24 | 0.45 | 0.44 | 3.9% | 0.0% | 0.82 | >141 |
| Fragment | TacoGFN | **100.0%** | -8.24 | -8.44 | 0.67 | 0.67 | 1.3% | 1.0% | 0.67 | **4** |
| 2D Reaction | RxnFlow | **100.0%** | -8.85 | -9.03 | 0.67 | 0.67 | 34.8% | 34.5% | 0.81 | **4** |
| 3D Reaction | 3DSynthFlow (low $\beta$) | 99.9% | -9.14 | -9.38 | 0.69 | 0.69 | **36.2%** | **37.0%** | 0.78 | 6 |
| | 3DSynthFlow (med $\beta$) | 99.9% | -9.30 | **-9.62** | 0.72 | 0.71 | 35.1% | 36.0% | 0.74 | 6 |
| | 3DSynthFlow (high $\beta$) | **100.0%** | **-9.42** | -9.61 | **0.73** | **0.73** | 36.1% | 36.0% | 0.69 | 6 |

thesizable drug set, including the autoregressive model **Pocket2Mol** (Peng et al., 2022), and diffusion-based methods: **TargetDiff** (Guan et al., 2023), **DiffSBDD** (Schneuing et al., 2024b), **DecompDiff** (Guan et al., 2024), and **MolCraft** (Qu et al., 2024). We further include comparisons with optimization-based approaches: the fragment-based method **TacoGFN** (Shen et al., 2024) and the reaction-based method **RxnFlow** (Seo et al., 2024). To ensure a fair evaluation of distribution learning-based methods, we use the docking proxy trained on CrossDocked training set. To balance exploration and exploitation during sampling, we vary the reward exponentiation parameter $\beta$ of **3DSynthFlow**: **low** $\beta$ (sampled from $\mathcal{U}(1, 64)$), **medium** $\beta$ ($\mathcal{U}(16, 64)$), and **high** $\beta$ ($\mathcal{U}(32, 64)$).

**Results.** As shown in Table. 4, 3DSynthFlow achieves improvements in pocket-conditional generation in synthesizability and docking scores. In particular, 3DSynthFlow-high $\beta$ attains an average docking score of -9.42 kcal/mol, outperforming RxnFlow (-8.85) and state-of-the-art diffusion-based methods like MolCRAFT-large (-9.25) and DecompDiff (-8.35). We attribute this improvement largely to our explicit consideration of 3D co-design in the reaction-based generation framework. However, due to the limited accuracy of the pre-trained proxy model for docking score reward, we find the improvement in docking score less significant than the pocket-specific setting.

3DSynthFlow surpasses the 2D reaction-based GFlowNets, TacoGFN and RxnFlow, while maintaining a highly comparable computational cost. We attribute this to a fundamental design difference: while the 2D baselines are conditioned on 1D sequence-based representations, 3DSynthFlow leverages the full 3D geometry of the binding site. This result indicates that 3D interaction modeling is essential, particularly for achieving robust performance across diverse targets.

SBDD methods such as MolCRAFT-large can attain strong binding affinity (-9.25) similar to 3DSynthFlow (-9.42), however their synthesis success rate is much lower (3.9% vs 36.1%). The main contribution of 3DSynthFlow is representing both the compositional nature for synthesis constraints and modeling 3D poses for binding.

# 7. Discussion

**Limitations.** Our synthesis-based action space based on arm and linker synthons does not yet explore certain nonlinear synthesis pathway such as ring-forming reactions. This synthon-based generation is chosen to prevent the leaving group atoms of intermediate states being lost or forming new rings – which would degrade the accuracy of pose prediction for intermediate states. However, this constraint can be substantially mitigated by expanding the chemical search space through incorporating a larger building block library. For example, Sadybekov et al. (2021) successfully achieved a notable hit rate of 33% with arm-and-linker design.

**Conclusion.** In this work, we introduce Compositional Generative Flows (CGFlow), a flexible generative framework for jointly modeling compositional structures and continuous states. We introduce a simple extension to the flow matching interpolation process for handling compositional state transition. CGFlow enables the integration of GFlowNets for efficient exploration of compositional state spaces with flow matching for continuous state modeling. We apply CGFlow to 3D molecule and synthesis pathway co-design and develop 3DSynthFlow, which achieves state-of-the-art performance on both LIT-PCBA and CrossDocked benchmark. Future work includes using a more expressive model for pose prediction and developing more application-specific methods using CGFlow.

## Impact Statement

In small-molecule drug discovery, it is crucial to design molecules that possess both high binding affinity and synthesizability. Despite the clear importance of these goals, efforts to achieve them simultaneously have been constrained by the distinct nature of each objective. To address this challenge, we propose a generative modeling framework for compositional and continuous data. Our approach enables the joint generation of 3D binding poses within protein binding site and their synthetic pathways, addressing key challenges in target-based drug discovery. By improving the integration of synthesis constraints and structural modeling, this work has the potential to accelerate the development of novel therapeutics, ultimately benefiting public health. At the same time, generative models for molecular design carry inherent risks, including the potential for misuse in designing harmful substances. To promote responsible use, we focus on applications that align with therapeutic discovery.

## Acknowledgements

This work was supported by Basic Science Research Programs through the National Research Foundation of Korea (NRF), grant-funded by the Ministry of Science and ICT (RS-2023-00257479). This research was also supported by the NSERC Discovery Grant.

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

# Appendix to:

# Compositional Flows for 3D Molecule and Synthesis Pathway Co-design

## A. CGFlow details

All code is provided in supplementary material.

### A.1. CGFlow recipe

We now summarize the key steps for implementing CGFlow for generative tasks which constructs compositional structures with continuous states.

1. Decompose the object into its compositional and continuous part (e.g., 2D structure and 3D coordinates) (See Sec. 3).
2. Define action space for additive composition steps.
3. Define a function for sampling action sequence for constructing the compositional structure (e.g. synthesis order).
4. Train State Flow model on existing datasets (e.g. 3D protein-ligand complexes) (Sec. 3.3.1).
5. Train the compositional flow model using GFlowNets-based Compositional Flow loss, if reward function $R(x)$ (docking score) is available, else use cross-entropy loss (Sec. 3.3.2).
6. Run sampling using trained state flow and compositional flow model (Algorithm 1).

### A.2. Sampling Algorithm

---

**Algorithm 1** Compositional Flow Sampling with CGFlow

---

**Require:** step size $\Delta t$, interval $\lambda$, time window $t_{\text{window}}$
1: **init** $t = 0, i = 0, \mathcal{C}_t = \mathcal{C}_0, \mathcal{S}_0 = \mathcal{S}_0, \hat{\mathcal{S}}_1^t = \mathcal{S}_0$
2: **while** $t < 1$ **do**
3:     $\boldsymbol{x}_t \leftarrow (\mathcal{C}_t, \mathcal{S}_t, \hat{\mathcal{S}}_1^t)$         ▷ *Use self-conditioning.*
4:     **if** $t \bmod \lambda = 0$ **then**
5:         $i \leftarrow i + 1$
6:         $\mathbf{C}^{(i)} \sim \pi_\theta(\boldsymbol{x}_t)$     ▷ **Compositional flow model:** *Sample the next compositional component $i$.*
7:         $\boldsymbol{S}_t^{(i)} \sim \mathcal{N}(0, 1)$     ▷ *Initialize new state values for component $i$ under the fixed seed*
8:         $\boldsymbol{x}_t \leftarrow T(\boldsymbol{x}_t, \mathbf{C}^{(i)}, \boldsymbol{S}_t^{(i)})$     ▷ *Transition with sampled composition and state value.*
9:         $t_{\text{gen}}^{(i)} \leftarrow t$
10:    **end if**
11:    $t_{\text{local}}^{(j)} \leftarrow \text{clip}\left(\frac{t - t_{\text{gen}}^{(j)}}{t_{\text{window}}}\right), \forall j \leq i$
12:    $\hat{\boldsymbol{S}}_1^{(j)} \leftarrow p_{1|t}^\theta(\boldsymbol{x}_t, t_{\text{local}}^{(j)}), \forall j \leq i$     ▷ **State flow model**: *Predict final state values.*
13:    $\boldsymbol{S}_{t+\Delta t}^{(j)} \leftarrow \boldsymbol{S}_t^{(j)} + (\hat{\boldsymbol{S}}_1^{(j)} - \boldsymbol{S}_t^{(j)}) \cdot \kappa^{(j)} \Delta t, \forall j \leq i$     ▷ *Step according to the predicted vector field.*
14:    $t \leftarrow t + \Delta t$
15: **end while**
16: **return** $\boldsymbol{x}_1$

---

### A.3. Alternative training objectives

There may be cases where reward function $R(x)$ is not available, or the goal is to model the data distribution instead. In these settings, the compositional flow model can adopt cross-entropy loss to maximize the log-likelihood of sampling $\mathbf{C}^{(\sigma_i)}$ at each step $i$, aligned with the valid generation order $\sigma$. The loss is:

$$\mathcal{L}_{\text{comp}} = -\mathbb{E}_{p_t(\boldsymbol{x}_t)} \sum_{i=1}^{n} \log \pi^\theta(\mathbf{C}^{(\sigma_i)} | \boldsymbol{x}_{t=t_{gen}^{(\sigma_i)}}), \tag{11}$$

where $\pi^\theta(\mathbf{C}^{(\sigma_i)} | \boldsymbol{x}_t)$ represents the policy model for generating the next compositional component $\mathbf{C}^{(\sigma_i)}$, conditioned on the entire object $\boldsymbol{x}_t$.

## B. Theoretical background

The object $x = (\mathcal{C}, \mathcal{S})$ is sequence data where $\mathcal{C} = (\boldsymbol{C}^{(i)})_{i=1}^n$ and $\mathcal{S} = (\boldsymbol{S}^{(i)})_{i=1}^n$. Therefore, we formulate the generative process as an auto-regressive process, i.e., $P_B(-|-) = 1.0$.

To train the compositional flow model with the trajectory balance (TB) objective, we must estimate forward transition probabilities along the trajectory $\tau = (\boldsymbol{x}_0 \to \boldsymbol{x}_{i\lambda} \to \cdots \to \boldsymbol{x}_{n\lambda} \to \boldsymbol{x}_1)$, where $\lambda$ is the time interval between successive components $\boldsymbol{C}^{(\cdot)}$, and $n < 1/\lambda$ is the total number of components.

Since the state flow model $p_{1|t}^\theta$ introduces randomness in the initial state $\boldsymbol{S}_0^{(i)} \sim \mathcal{N}(0, \sigma^2)$, estimating the forward transition probability involves integrating over this noise distribution as follows:

$$
\begin{aligned}
P_F\left(\boldsymbol{x}_{(i+1)\lambda}\Big|\boldsymbol{x}_{i\lambda}, \hat{\boldsymbol{x}}_1^t; \phi, p_{1|t}^\theta\right) &= P_F\left(\boldsymbol{C}^{(i)}\Big|\boldsymbol{x}_{i\lambda}, \hat{\boldsymbol{x}}_1^t; \phi\right) \int P\left(\boldsymbol{x}_{(i+1)\lambda}\Big|T(\boldsymbol{x}_{i\lambda}, \boldsymbol{C}^{(i)}, \boldsymbol{S}_0^{(i)}); p_{1|t}^\theta\right) p(\boldsymbol{S}_0^{(i)})d\boldsymbol{S}_0^{(i)} \\
&= P_F\left(\boldsymbol{C}^{(i)}\Big|\boldsymbol{x}_{i\lambda}, \hat{\boldsymbol{x}}_1^t; \phi\right) \int \delta\left(\Phi^\theta\left(T(\boldsymbol{x}_{i\lambda}, \boldsymbol{C}^{(i)}, \boldsymbol{S}_0^{(i)}), \lambda, \Delta t\right) = \boldsymbol{x}_{(i+1)\lambda}\right) p(\boldsymbol{S}_0^{(i)})d\boldsymbol{S}_0^{(i)},
\end{aligned}
\tag{12}
$$

where $\Phi^\theta$ is the ODE solver of the state flow model $p_{1|t}^\theta$. This integration induces a Dirac delta term, making direct probability estimation challenging.

**Proposition B.1.** *Let the initial state $\boldsymbol{S}_0^{(i)}$ be fixed[§]. Then, the object $\boldsymbol{x}_t = (\mathcal{C}_t, \mathcal{S}_t)$ is uniquely determined by a given trajectory $\tau$ sampled from the compositional flow model $\pi^\phi$.*

*Proof.* When $\boldsymbol{S}_0^{(i)}$ is fixed, the ODE solver fully specifies $\mathcal{S}_{(i+1)\lambda}$ based on $\mathcal{S}_{i\lambda}$, the self-conditioning $\hat{\boldsymbol{x}}_1^t$, and the sampled component $\boldsymbol{C}^{(i)}$. Consequently, the sequence $(\boldsymbol{C}^{(1)}, \ldots, \boldsymbol{C}^{(k(t))})$ directly determines $\mathcal{S}_t$. Given that the generative progress is auto-regressive, there is a one-to-one correspondence between $\boldsymbol{x}_t$ and $(\boldsymbol{C}^{(1)}, \ldots, \boldsymbol{C}^{(k(t))})$ for every $t$.

Building on this determinism, we simplify the forward transition probability $P_F(\boldsymbol{x}_{(i+1)\lambda}|\boldsymbol{x}_{i\lambda;\phi;p_{1|t}^\theta})$ as:

$$
P_F\left(\boldsymbol{x}_{(i+1)\lambda}\Big|\boldsymbol{x}_{i\lambda}, \hat{\boldsymbol{x}}_1^t; \phi, p_{1|t}^\theta\right) = P_F\left(\boldsymbol{C}^{(i)}\Big|\boldsymbol{x}_{i\lambda}, \hat{\boldsymbol{x}}_1^t; \phi\right)
\tag{13}
$$

This allows us to write the TB objective in a more tractable form:

$$
\begin{aligned}
\mathcal{L}_{\text{TB}}(\tau) &= \left(\log \frac{Z_\phi \prod_{i=0}^{n-1} P_F(\boldsymbol{x}_{(i+1)\lambda}|\boldsymbol{x}_{i\lambda}; \phi, p_{1|t}^\theta)}{R(\boldsymbol{x}_1)\prod_{i=0}^{n-1} P_B(\boldsymbol{x}_{i\lambda}|\boldsymbol{x}_{(i+1)\lambda}; \phi, p_{1|t}^\theta)} \times \frac{\cancel{P(\boldsymbol{x}_1|\boldsymbol{x}_{n\lambda}; p_{1|t}^\theta)}}{\cancel{P(\boldsymbol{x}_{n\lambda}|\boldsymbol{x}_1; p_{1|t}^\theta)}}\right)^2 \\
&= \left(\log \frac{Z_\phi \prod_{i=0}^{n-1} P_F(\boldsymbol{C}^{(i)}|\boldsymbol{x}_{i\lambda}; \phi)}{R(\boldsymbol{x}_1)}\right)^2.
\end{aligned}
\tag{14}
$$

By enforcing trajectory balance under these conditions, we ensure that the likelihood of sampling a final object $\boldsymbol{x}_1$ is proportional to its reward $R(\boldsymbol{x}_1)$, while sidestepping the complexities introduced by the continuous noise integration.

## C. Extended related works

**Sequential diffusion for molecular generation** Peng et al. (2023); Ghorbani et al. (2023); Li et al. (2024) sequentially generate molecules fragment by fragment using diffusion models. These methods use a separate diffusion process to generate each 3D fragment graph, with atomic positions fixed post-generation. They also lack the ability to enforce compositional synthesis constraints during the generation process. Instead, 3DSynthFlow formulates a joint flow process: state flow refines all atomic positions throughout, mitigating the issue with cascading error in position prediction; composition flow sequentially constructs the synthesis pathway, effectively enforcing the synthesis constraint.

---

[§]To minimize the influence of the fixed initial state value, we sample the initial state from the noise distribution using the manual random seed which is equal to the size of the current object, such as the number of atoms.

**Structure-based drug design** We categorize structure-based drug design (SBDD) in two main categories: pocket-specific and pocket-conditional following Seo et al. (2024).

The first approach optimizes docking scores for a target. Methods include evolutionary algorithms (Reidenbach, 2024), reinforcement learning (RL) (Zhavoronkov et al., 2019), and GFlowNets (Bengio et al., 2021; Pandey et al., 2025; Koziarski et al., 2024). The drawback of this approach is that each pocket must be optimized individually, which can restrict scalability.

In contrast, the pocket-conditional generation approach produces molecules tailored to any given pocket without the need for extra training. This strategy leverages distribution-based generative models (Ragoza et al., 2022; Peng et al., 2022; Guan et al., 2024; Schneuing et al., 2024b; Qu et al., 2024) that are trained on protein-ligand complex datasets to learn the distribution of ligands suitable for different pockets. Recently, Shen et al. (2024); Seo et al. (2024) adopted pocket-conditioned policy for GFlowNets that generates samples from reward-biased distributions in a zero-shot setting.

# D. 3DSynthFlow details

## D.1. Action space

Following Cretu et al. (2024); Koziarski et al. (2024); Seo et al. (2024), we treat chemical reactions as forward transitions and synthetic pathways as trajectories for molecular generation.

Compared to previous methods, we represent a building block as *synthon*, which is not a complete molecule. The synthons can be connected at the *attachment point* according to the pre-defined connection rules, i.e., reactions $\mathcal{R}$ (See Fig. 4). To prevent a generation trajectory terminating at an incomplete structure, we define two types of synthon according to Liu et al. (2017): *brick* is the synthon including one attachment point, and *linker* is the synthon including two attachment points.

*Figure 6.* Examples of brick and linker synthons.

We define $\mathcal{B}$ as the entire synthon set, $\mathcal{B}' \subseteq \mathcal{B}$ as the brick synthon set, and $\mathcal{B}_r \subseteq \mathcal{B}$ as the allowable synthon set for reaction $r \in \mathcal{R}$.

At the initial state $\mathcal{C}_0$, the model always selects `FirstSynthon`, which samples a brick synthon $b$ from the entire brick synthon set $\mathcal{B}'$ to serve as the starting molecule. For subsequent states $\mathcal{C}_t$, the model chooses `AddSynthon`, which firstly identifies the available reaction set $\mathcal{R}(\mathcal{C}_t) \subseteq \mathcal{R}$ and then samples the synthon from the available synthon set $\cup_{r \in \mathcal{R}(\mathcal{C}_t)} \mathcal{B}_r$. If the brick synthon is selected, the trajectory is terminated. When the trajectory reaches the maximum length, the model always selects a brick synthon.

In summary, the allowable action space $\mathcal{A}(\mathcal{C})$ for a given compositional state $\mathcal{C}$ is:

$$\mathcal{A}(\mathcal{C}) = \begin{cases} \mathcal{B}' & \text{if } t = 1, \\ \cup_{r \in \mathcal{R}(\mathcal{C})} \mathcal{B}_r & \text{otherwise,} \end{cases} \tag{15}$$

## D.2. 3DSynthFlow Model architecture

### D.2.1. STATE FLOW MODEL

To model state flow for predicting ligand docking poses in 3DSynthFlow, we extend the *Semla* architecture introduced by Irwin et al. (2024). *Semla* is a scalable, E(3)-equivariant model originally designed for 3D molecular generation; We refer reader to the original paper for full details for *SemlaFlow*. We make the following modifications to adapt *Semla* for 3D protein-ligand modelling:

**Hierarchical pocket encoding:** Inspired by AlphaFold3 (Abramson et al., 2024), which models protein at both atom level and token level, we propose a hierarchical encoding strategy *HierSemla* which first encodes each residues at atom level and aggregates atomic representation to residue level. More specifically, we introduce a *Residue Encoder* based on the

Semla architecture, and apply it independently to encode each of its atom type, connections and position first. Then we perform mean pooling over the invariant and equivariant features within each residue, obtaining residue-level embedding $(\boldsymbol{h}_i^{(\mathrm{pro})}, \boldsymbol{x}_i^{(\mathrm{pro})})$. This hierarchical design enables all-atom resolution modeling of the protein pocket while significantly improving memory efficiency: by restricting full attention to within-residue interactions, we avoid the $\mathcal{O}(N^2)$ memory cost of full atom-level attention across the entire protein, which can contain up to 2000 atoms.

**Synthon attachment point information:** In addition to the original atomic input features (atomic type and charge), we further provide the boolean feature IS_ATTACHMENT_POINT indicating whether this atom is the attachment point for a new synthon. Intuitively, this hints the model where the future synthon will be added in next reaction and bias the model to leave space for the future synthon in the intermediate state pose prediction. We find this additional feature lead to better pose prediction performance in practice.

**Protein-ligand attention layer:** The protein residue embeddings $(\boldsymbol{h}_i^{(\mathrm{pro})}, \boldsymbol{x}_i^{(\mathrm{pro})})$, along with their original positions $\boldsymbol{x}_i^{(\mathrm{pro})}$, are combined with the ligand atom embeddings $(\boldsymbol{h}_j^{(\mathrm{lig})}, \boldsymbol{x}_j^{(\mathrm{lig})})$ for pairwise message passing at each modified *Semla* layer $(i, j) \in V \times P$, defined as:

$$\left(\boldsymbol{m}_{i,j}^{\mathrm{inv}},\ \boldsymbol{m}_{i,j}^{\mathrm{equi}}\right) = \Omega_\theta^{(pro-lig)}\left(\tilde{\boldsymbol{h}}_i,\ \tilde{\boldsymbol{h}}_j,\ \tilde{\boldsymbol{x}}_i \cdot \tilde{\boldsymbol{x}}_j\right), \tag{16}$$

where $\tilde{\boldsymbol{h}}_i$ and $\tilde{\boldsymbol{x}}_i$ are normalized linearly projected features. The protein-ligand messages are aggregated with the original ligand-ligand messages and used for attention-based updates. This formulation enables message exchange between residue nodes and atom nodes, ensuring that the model effectively learns protein-ligand interactions critical to binding.

**Modification to prior:** Since the atom type and bond connections are generated by the compositional flow model instead of the state flow model, therefore they remain fixed throughout the process. This means we cannot use techniques such as equivariant-OT to reduce the transport cost (Klein et al., 2023; Song et al., 2023), since they assume interchangeability between newly initialized nodes. We have also tried using harmonic prior from Jing et al. (2023); Stärk et al. (2024) to initialize position with bonding information as prior. We observed improved pose prediction with Gaussian noise during initial training, but no advantage of the harmonic prior over the Gaussian prior at convergence.

### D.2.2. COMPOSITIONAL FLOW MODEL

We adopt a model architecture inspired by Bengio et al. (2021); Cretu et al. (2024); Seo et al. (2024), using a graph transformer (Yun et al., 2022) as the backbone $f_\theta$, and a multi-layer perceptron (MLP) $g_\theta$ for action embedding. The graph embedding dimension is $d_1$ and the synthon embedding dimension is $d_2$. The GFlowNet condition such as temperature $\beta$ or multi-objective weights are encoded in condition vector $c$.

To capture spatial relationships between the protein and ligand, the model uses a 3D protein-ligand complex graph for each state. The protein is represented as a fully-connected residual graph, where each node corresponds to a $C_\alpha$ atom. The ligand is also connected to all protein nodes. All pairwise distances between nodes are encoded in the graph edges, enabling the model to reason about the 3D structure. We represent each state $s$ as an input of GFlowNet.

**Initial synthon selection.** For the initial state $s = s_0$, the model always selects FirstSynthon to sample brick synthon $b \in \mathcal{B}'$ for the starting molecule with $\mathrm{MLP}_{\texttt{AddSynthon}} : \mathbb{R}^{d^1} \to \mathbb{R}^{d^2}$.

$$F_\theta(s_0, b, c) = \mathrm{MLP}_{\texttt{FirstSynthon}}(f_\theta(s_0, c)) \odot g_\theta(b) \tag{17}$$

**Adding synthon selection.** For the later states $s \neq s_0$, the model selects AddSynthon to sample allowable brick or linker synthon $b \in \cup_{r \in \mathcal{R}(s)} \mathcal{B}_r$ with $\mathrm{MLP}_{\texttt{AddSynthon}} : \mathbb{R}^{d^1} \to \mathbb{R}^{d^2}$. We note that the reaction type information is included in synthon embedding:

$$F_\theta(s, (r, b), c) = \mathrm{MLP}_{\texttt{AddSynthon}}(f_\theta(s, c)) \odot g_\theta(r, b). \tag{18}$$

**Synthon masking.** The state flow model is trained on the data distribution of active ligands. This may sometimes not work well for out-of-distribution molecules that are much larger than the training data. Therefore, we restrict the actions which make the state to have more than 40 atoms. This process can be performed simply in a synthon-based action space.

# E. Experimental details

## E.1. Training details

### E.1.1. 3DSynthFlow hyperparameters

*Table 5.* **Default hyperparameters** used for compositional structure.

| Hyperparameters | Values |
|---|---|
| Time per action $\lambda$ | 0.3 |
| Interpolation window $t_{window}$ | 1.0 |
| Maximum decomposed part | 3 |
| Minimum decomposed fragment size | 5 |
| Minimum trajectory length | 2 (minimum reaction steps: 1) |
| Maximum trajectory length | 3 (maximum reaction steps: 2) |

Here we define the hyperparameters used for 3DSynthFlow:

1. **Time per action** ($\lambda$): Defines the time interval between adding each additional synthon.

2. **Interpolation window** ($t_{window}$): Specifies the fixed time window that affects the noise scheduling of the continuous states of each component.

3. **Maximum decomposed part**: Determines the maximum number of synthons a molecule can be decomposed into, preventing molecules from being associated with excessive synthesis steps.

4. **Minimum decomposed fragment size**: Specifies the minimum number of atoms that each synthon product must contain when a molecule is decomposed according to reaction rules. This ensures that synthons are of realistic size when decomposing CrossDocked ligands for training a pose prediction model.

5. **Minimum trajectory length**: Defines the minimum trajectory length for sampling composition steps in 3DSynthFlow.

6. **Maximum trajectory length**: Specifies the maximum trajectory length that can be reached.

We further exam the effect of performance for different setting of time scheduling, varying $\lambda$ and $t_{window}$ in Appendix F.3.

*Table 6.* **Default hyperparameters** used for State flow model.

| Hyperparameters | Values |
|---|---|
| Number of protein layers | 4 |
| Number of ligand layers | 8 |
| Noise prior | Gaussian |
| Time distribution | Beta(1.0, 1.0) |

For parameterizing the state flow model for pose prediction, we use the same hyperparameter from the official repository [¶]. We show the additional hyperparameters in Table. 6 and provide an explanation for each below:

1. **Number of protein layers**: the number of layers with protein-protein message passing.

2. **Number of ligand layers**: the number of layers with protein-ligand message passing.

3. **Noise prior**: Defines the noise distribution which initial state values are drawn from.

4. **Time distribution**: Defines the time distribution

---

[¶]Follow SemlaFlow's hyperparameter settings in `train.py` at https://github.com/rssrwn/semla-flow/blob/main/semlaflow/train.py

**Composition flow model**    In this work, we set most parameters to the default values[‖] for all GFlowNet baselines (Some of the parameters are in Table. 7). However, since 3DSynthFlow is built from RxnFlow, it follows some important parameters of RxnFlow except for maximum trajectory length and state embedding size. We note that our state embedding dimension is smaller than other GFlowNet baselines.

*Table 7.* Default hyperparameters used in all GFlowNets. The settings are from `seh_frag.py` (Bengio et al., 2021)

| Hyperparameters | Values |
| --- | --- |
| GFN temperature $\beta$ | Uniform(0, 64) |
| Sampling tau (EMA factor) | 0.9 |
| Learning rate ($Z$) | $10^{-3}$ |
| Learning rate ($P_F$) | $10^{-4}$ |
| State embedding dimension | 64 |

*Table 8.* Specific hyperparameters used in 3DSynthFlow training. The parameters are from RxnFlow (Seo et al., 2024), except for the maximum trajectory length (4 for RxnFlow) and state embedding size (128 for RxnFlow and other GFlowNets).

| Hyperparameters | Values |
| --- | --- |
| Minimum trajectory length | 2 (minimum reaction steps: 1) |
| Maximum trajectory length | 3 (maximum reaction steps: 2) |
| State embedding dimension | 64 |
| Action embedding dimension | 64 |
| Action space subsampling ratio | 1% |
| Train random action probability | 5% |

For action space subsampling, we randomly subsample 1% actions for `FirstSynthon` and each bi-molecular reaction template $r \in \mathcal{R}_2$. However, for bi-molecular reactions with small possible reactant block sets $\mathcal{B}_r \in \mathcal{B}$, the memory benefit from the action space subsampling is small while a variance penalty is large. Therefore, we set the minimum subsampling size to 100 for each bi-molecular reaction, and the action space subsampling is not performed when the number of actions is smaller than 100.

The number of actions for each action type is imbalanced, and the number of reactant blocks ($\mathcal{B}_r$) for each bi-molecular reaction template $r$ is also imbalanced. This can make some rare action categories not being sampled during training. We empirically found that `ReactBi` action were only sampled during 20,000 iterations (1.28M samples) in a toy experiment that uses one bi-molecular reaction template and 10,000 building blocks in some random seeds. Therefore, we set the random action probability as the default of 5%, and the model uniformly samples each action category in the random action sampling. This prevents incorrect predictions by ensuring that the model experiences trajectories including rare actions. We note that this random selection is only performed during model training.

### E.1.2. DETAILS OF STATE FLOW MODEL TRAINING

To expose the state flow model to realistic partial ligand structures, we decompose each CrossDocked ligand into up to three fragments using 38 bimolecular Enamine synthesis protocols, defined by reaction SMARTS patterns. For each molecule, we randomly select a fragment ordering and sample a time step $t$, so the model observes the molecule at varying stages of assembly (e.g., only fragment A at $t = 1$, fragments A and B at $t = 2$, etc.). This mimics the compositional assembly process used during generation and teaches the model to predict physically plausible docking conformations for incomplete ligands.

Importantly, the decomposition is not intended to yield commercially purchasable Enamine synthons. Instead, the use of Enamine protocols ensures the resulting fragments are chemically meaningful and synthetically relevant, even if they do not match existing catalog synthons. This allows us to use any CrossDocked molecule for training while staying aligned with the synthetic priors used at generation time.

Across the CrossDocked dataset, we find that over 93% of molecules can be decomposed into at least two fragments using

---

[‖]Follow default hyperparameter settings in `seh_frag.py` and `seh_frag_moo.py` at https://github.com/recursionpharma/gflownet/blob/trunk/src/gflownet/tasks

this protocol, and over 87% can be decomposed into three. This high decomposition rate enables robust learning from partially constructed molecules and ensures strong coverage of fragment combinations encountered during inference.

### E.1.3. TRAINING TIME AND COMPUTATIONAL EFFICIENCY

The state flow model (i.e., the pocket-conditional pose predictor) was trained for 30 epochs on 8 A4000 GPUs (16GB), requiring a total of 26 hours. Since the state flow model is trained on the CrossDocked dataset and is reused across different test pockets, it incurs only a one-time computational cost. We plan to release the pretrained weights, so that users will need to train only the composition flow model tailored to their custom reward function and target.

In contrast, the composition flow model is trained for 1,000 iterations under 20 flow matching steps. Training with GPU-accelerated docking (UniDock; balance mode) takes between 7 and 12 hours (depending on the target) on a single A4000 GPU (16GB), making the computational requirement accessible for most practical drug discovery campaigns. Moreover, the composition flow model can be trained in a pocket-conditioned manner to enable zero-shot molecule sampling for any target pocket, thereby converting the training into a one-time cost in this setting.

*Table 9.* Summary of Training Time for 3DSynthFlow Components.

| Model Component | Hardware | Batch Size | Number of Iterations | Training Time |
|---|---|---|---|---|
| State Flow Model | $8 \times$ A4000 (16GB) | *dynamic* | 30 epochs | 26 hours |
| Composition Flow Model | $1 \times$ A4000 (16GB) | 64 | 1,000 steps | 7–10 hours |

### E.2. Reward function

To train GFlowNet models, we employ the same reward function proposed in RxnFlow (Seo et al., 2024) and TacoGFN (Shen et al., 2024).

**Pocket-specific optimization** . To optimize both QED andd Vina docking score, we set the reward function as:

$$R(x) = w_1 \text{QED}(x) + w_2 \widehat{\text{Vina}}(x), \tag{19}$$

where $\widehat{\text{Vina}}(x)$ is a normalized docking score. The parameters $w_1$ and $w_2$ serve as conditions of multi-objective GFlowNets (Jain et al., 2023b), and are set to 0.5 for non-GFlowNet baselines.

**Pocket-conditional optimization** . We use the the modified reward function proposed by Shen et al. (2024). According to Seo et al. (2024), we remove SA score term (Ertl & Schuffenhauer, 2009) from the reward functions since 3DSynthFlow explicitly generate synthetic pathway:

$$r_{\text{affinity}}(x) = \begin{cases} 0 & \text{if } 0 \leq \text{Proxy}(x) \\ -0.04 \times \text{Proxy}(x) & \text{if } -8 \leq \text{Proxy}(x) \leq 0 \\ -0.2 \times \text{Proxy}(x) - 1.28 & \text{if } -13 \leq \text{Proxy}(x) \leq -8 \\ 1.32 & \text{if } \text{Proxy}(x) \leq -13 \end{cases}$$

$$r_{\text{QED}}(x) = \begin{cases} \text{QED}(x)/0.7 & \text{if } \text{QED}(x) \leq 0.7 \\ 1 & \text{otherwise} \end{cases}$$

$$r_{\text{SA}}(x) = \begin{cases} \widehat{\text{SA}}(x)/0.8 & \text{if } \widehat{\text{SA}}(x) \leq 0.8 \\ 1 & \text{otherwise} \end{cases}$$

$$\text{TacoGFN-Reward}(x) = \frac{r_{\text{affinity}}(x) \times r_{\text{QED}}(x) \times r_{\text{SA}}(x)}{\sqrt[3]{\text{HeavyAtomCounts}(x)}} \tag{20}$$

$$\text{RxnFlow-Reward}(x) = \frac{r_{\text{affinity}}(x) \times r_{\text{QED}}(x)}{\sqrt[3]{\text{HeavyAtomCounts}(x)}} \tag{21}$$

### E.3. Datasets

#### E.3.1. LIT-PCBA POCKETS

Table. 10 describes the protein information used in pocket-specific optimization with UniDock, which is performed on Sec. 6. We follow the same procedure used in pocket extraction for the CrossDock dataset: taking all residue of the protein within 10 Å radius to the reference ligand as the binding pocket.

*Table 10.* **The basic target information** of the LIT-PCBA dataset and PDB entry used in this work.

| Target | PDB Id | Target name |
|--------|--------|-------------|
| ADRB2 | 4ldo | Beta2 adrenoceptor |
| ALDH1 | 5l2m | Aldehyde dehydrogenase 1 |
| ESR_ago | 2p15 | Estrogen receptor $\alpha$ with agonist |
| ESR_antago | 2iok | Estrogen receptor $\alpha$ with antagonist |
| FEN1 | 5fv7 | FLAP Endonuclease 1 |
| GBA | 2v3d | Acid Beta-Glucocerebrosidase |
| IDH1 | 4umx | Isocitrate dehydrogenase 1 |
| KAT2A | 5h86 | Histone acetyltransferase KAT2A |
| MAPK1 | 4zzn | Mitogen-activated protein kinase 1 |
| MTORC1 | 4dri | PPIase domain of FKBP51, Rapamycin |
| OPRK1 | 6b73 | Kappa opioid receptor |
| PKM2 | 4jpg | Pyruvate kinase muscle isoform M1/M2 |
| PPARG | 5y2t | Peroxisome proliferator-activated receptor $\gamma$ |
| TP53 | 3zme | Cellular tumor antigen p53 |
| VDR | 3a2i | Vitamin D receptor |

#### E.3.2. CROSSDOCKED2020

We train the State flow model of 3DSynthFlow on the commonly used **CrossDocked** dataset (Francoeur et al., 2020). We apply the splitting and processing protocol on the CrossDocked dataset to obtain the same training split of protein-ligand pairs as previous methods (Luo et al., 2021; Peng et al., 2022). We use 99,000 complexes as the training set and remaining 1,000 complexes as the validation set. Since we co-design 3D binding pose and synthesis pathway, unlike the pervious 2D-based methods, we can leverage this dataset for training the auxiliary State flow pose prediction model.

### E.4. Baselines

**SynNet, BBAR** We reused the values reported in Seo et al. (2024).

**FragGFN, RGFN, RxnFlow.** All GFlowNet baselines share the same training parameters under the multi-objective GFlowNet (Jain et al., 2023b) setting. We also reused the values reported in Seo et al. (2024).

**SynFlowNet.** There are two versions for SynFlowNet (Cretu et al., 2024): 2024.3 and 2024.8. For version 2024.3, we reused the values in Seo et al. (2024). For version 2024.8, we followed the processes and settings according to the original paper and official code repository [**]. To construct the action space, we randomly selected 10,000 building blocks from Enamine Global Stock (v2025.01.11) with 105 reaction templates. We trained SynFlowNet using backward policy learning maximum likelihood, maximum trajectory length to 3, and action embedding with Morgan fingerprint (Morgan, 1965). Finally, we set the training parameters used in other GFlowNet baselines (See Table. 7). The training code and data used in the benchmark study are included in the supplementary materials.

## F. Additional results

### F.1. Reward computation without full docking

The most computational and time-intensive process in the training process is often docking the generated candidates with molecular docking for evaluation. Glide, a standard docking software in the industry, takes around 6 minutes per compound

---

[**]Follow SynFlowNet's hyperparameter settings in `reactions_task.py` at `https://github.com/mirunacrt/synflownet/tree/46a4acabd2255eb964c317ffbb86b743a13a4685`

*Table 11.* Ablation study on reward setting for Vina docking scores. Evaluated on the ADH1 pocket. Results are averaged over 3 runs.

| Method | Reward setting | Docking score ($\downarrow$) |
|---|---|---|
| RxnFlow | Full docking | -11.26 ($\pm$ 0.07) |
| **3DSynthFlow** | Full docking | -11.82 ($\pm$ 0.07) |
| **3DSynthFlow** | Local opt. | -11.44 ($\pm$ 0.06) |
| **3DSynthFlow**$_{\text{finetuned}}$ | Local opt. | -11.62 ($\pm$ 0.02) |

in its most accurate setting (Friesner et al., 2004). This constraint necessitates either reducing the number of Oracle queries or trading speed for less accurate docking settings. Docking scores are typically computed via **Full docking**, which performs a full search for the optimal binding pose at a high computational cost. However, it can also be computed via **Local optimization** for significantly faster computation, if an existing binding pose is available. [††]

In this setting, we investigate directly using the final predicted pose from 3DSynthFlow to compute reward using local optimization compared to using the full docking score. In the 3DSynthFlow $_{\text{finetuned}}$ setting for local opt., we first finetune the pose predictor on re-docked poses from the first 9,600 sampled molecules to improve binding pose prediction, thereby enabling accurate local optimization docking scores.

From Table. 11, we see that regardless of the scoring function used or whether we perform fine tuning, 3DSynthFlow outperformed the best 2D baseline RxnFlow, confirming the benefit of 3D structure co-design. Interestingly, 3DSynthFlow using local opt. do not perform as well as 3DSynthFlow trained with signals from full docking in both settings. While fine tuning the pose prediction module helps, the gap between full docking and local opt. is not fully closed. Empirically, we find the poses predicted from 3DSynthFlow sometimes contain steric clashes, leading to inaccurate estimation of the reward signal. Accordingly, improving the pose prediction module emerges as a key next step to reduce steric clashes and enhance the accuracy of local optimization docking scores, as disccussed in Sec. 7.

By co-designing binding pose and molecule, 3DSynthFlow using local opt. can bypass the computational burden of full docking which 2D generative methods are subjected to, while surpassing their performance. This is a key advantage for 3DSynthFlow in cases where accurate full docking is prohibitively expensive for a large number of ligands.

## F.2. Effect of Flow Matching Steps

*Table 12.* Effect of flow matching steps on performance for the ALDH1 target with the Vina reward. We report both the average docking score (Avg Vina) over all generated molecules and the top 100 docking scores (Top 100 Vina), with lower values indicating better binding. Training time is reported in seconds per iteration, and sampling time is reported in seconds per molecule.

| Flow Matching Steps | Avg Vina ($\downarrow$) | Top 100 Vina ($\downarrow$) | Training Time (sec/iter) | Sampling Time (sec/mol) |
|---|---|---|---|---|
| 20 | $-8.55 \pm 0.18$ | $-12.86 \pm 0.22$ | 16 | 0.058 |
| 40 | $-8.57 \pm 0.13$ | $-13.03 \pm 0.27$ | 22 | 0.086 |
| 60 | $-8.84 \pm 0.14$ | $-13.50 \pm 0.24$ | 24 | 0.115 |
| 80 | $-8.47 \pm 0.18$ | $-13.15 \pm 0.16$ | 27 | 0.153 |

We further analyzed how the number of flow matching steps impacts performance using the ALDH1 target with the Vina reward. As shown in Table 12, performance slightly improves with increasing flow matching steps and appears to saturate around 40–60 steps. This marginal improvement may stem from the fact that the pose prediction module's primary role is to provide a spatial context between intermediate molecules and the pocket; hence, extremely precise pose predictions have limited additional impact on model decisions.

## F.3. Ablation study on time scheduling of 3DSynthFlow

We conducted an ablation study to assess the effect of time scheduling in state flow training. Specifically, we compared three scheduling strategies:

- **No overlap**: strictly autoregressive denoising - each synthon denoised after the previous one is completed. ($\lambda = 0.33, t_{window} = 0.33$).

---

[††]Scoring a predicted pose using the local-optimization setting in AutoDock Vina is 7× faster than performing full docking. This speedup is even greater for more accurate docking programs, since the vast majority of time is spent on pose searching rather than scoring.

- **Overlapping**: partial overlap of synthon denoising ($\lambda = 0.3, t_{window} = 0.4$).

- **Till end**: all synthons are denoised jointly until $t = 1$ ($\lambda = 0.33, t_{window} = 1.0$).

To highlight the impact of noise scheduling on the final pose, we report the average local-optimized Vina docking scores across different training iterations for the ALDH1 target over 3 seeds:

| # of mol explored | 10,000 | 20,000 | 30,000 |
|---|---|---|---|
| No overlap | $-5.68 \pm 0.29$ | $-6.33 \pm 0.26$ | $-7.02 \pm 0.34$ |
| Overlapping | $-6.28 \pm 0.22$ | $-7.28 \pm 0.21$ | $-7.22 \pm 0.12$ |
| Till end | $-7.15 \pm 0.40$ | $-7.60 \pm 0.29$ | $-7.79 \pm 0.12$ |

*Table 13.* Ablation study on time scheduling for state flow training on the ALDH1 target.

We find 3DSynthFlow 's **Till end** strategy, where all positions are refined until time $t = 1$, clearly outperforms conventional autoregressive approaches - **No overlap**, adopted in previous works (See Appendix C). Our framework enables tuning of hyperparameter $\lambda$ and $t_{window}$ to accommodate varying data problems.

## F.4. State Space Size Estimation

We estimate the sample space size based on the number of synthetic steps: $10^{11}$ molecules with a single reaction step, $10^{17}$ molecules with two reaction steps, and $10^{23}$ molecules with three reaction steps. In our experiments, we employed up to two reaction steps according to Enamine REAL. The resulting state space size is comparable to RGFN (up to 4 steps with 8,350 blocks) and SynFlowNet (up to 3 steps with 200k blocks).

To assess the breadth of building block (BB) exploration, we analyzed the number of unique BBs encountered during training across the first five LIT-PCBA targets. Our model explored an average of approximately 55,000 unique BBs within 1,000 training iterations using a batch size of 64. This indicates significantly broader exploration compared to SynFlowNet, which reported around 15,000 unique BBs over 8,000 iterations with a batch size of 8.

*Table 14.* Number of unique building blocks explored during training across 5 LIT-PCBA targets.

| Target | ADRB2 | ALDH1 | ESR_ago | ESR_antago | FEN1 |
|---|---|---|---|---|---|
| **Number of Unique BBs** | $45520 \pm 7876$ | $48644 \pm 1983$ | $55211 \pm 5611$ | $58097 \pm 8529$ | $69400 \pm 5259$ |

## F.5. PoseCheck protein-ligand interactions

*Table 15.* **PoseCheck protein-ligand interactions.** We report two versions of SynFlowNet (v2024.05[a] and v2024.10[b]). Averages and standard deviations over 3 runs, for the top 100 diverse modes per LIT-PCBA pocket. The best results in each column are in **bold**.

| Method | PoseCheck Metrics ($\uparrow$) | | | | |
|---|---|---|---|---|---|
| | H-Bond Acceptors | H-Bond Donors | Van der Waals | Hydrophobic | Sum |
| SynFlowNet[a] | 0.22 ($\pm$ 0.01) | 0.11 ($\pm$ 0.01) | 9.31 ($\pm$ 0.05) | 10.41 ($\pm$ 0.05) | 20.05 |
| SynFlowNet[b] | 0.22 ($\pm$ 0.03) | 0.10 ($\pm$ 0.01) | 8.38 ($\pm$ 0.05) | 9.44 ($\pm$ 0.06) | 18.14 |
| RGFN | 0.19 ($\pm$ 0.01) | 0.11 ($\pm$ 0.01) | 9.19 ($\pm$ 0.28) | 10.25 ($\pm$ 0.12) | 19.73 |
| RxnFlow | 0.22 ($\pm$ 0.00) | 0.10 ($\pm$ 0.01) | 9.63 ($\pm$ 0.09) | 10.67 ($\pm$ 0.10) | 20.62 |
| **3DSynthFlow** | **0.33** ($\pm$ 0.06) | **0.17** ($\pm$ 0.03) | **11.04** ($\pm$ 0.37) | **10.79** ($\pm$ 0.21) | **22.32** |

Lastly, we evaluate the protein-ligand interactions of the top 100 generated molecules using PoseCheck (Harris et al., 2023). Since baselines do not generate 3D poses, we assess all methods using their redocked structures for consistency. PoseCheck evaluates four key protein-ligand interactions: H-bond acceptors, H-bond donors, van der Waals contacts, and hydrophobic effects.

Hydrogen bonds (H-bonds) are the most important specific interactions in protein-ligand recognition (Bissantz et al., 2010) and require precise geometric alignment to form (Brown, 1976). Unlike hydrophobic and van der Waals interactions, which

are non-directional and broadly applicable, H-bonds require strict atomic alignment, reinforcing the need for accurate 3D molecular modeling.

Notably, 3DSynthFlow achieves the highest H-bond acceptor and donor counts, outperforming all baselines. This improvement suggests that co-designing 3D structure and synthesis pathways enhances the geometric alignment of polar functional groups, leading to more stable and specific protein-ligand interactions. In addition, 3DSynthFlow-generated molecules also result in more hydrophobic and van der Waals interactions to baselines, as shown in Table. 15, further enhancing binding stability.

### F.6. Full sampling efficiency results

*Table 16.* Average number of diverse modes discovered versus the number of molecules explored. The best results are in bold. Results are computed over 2 seeds.

| Target | Method | Number of Molecules Explored | | | | |
| | | 1,000 | 5,000 | 10,000 | 30,000 | 64,000 |
|---|---|---|---|---|---|---|
| ADRB2 | RxnFlow | 3.0 (±1.4) | 21.5 (±4.9) | 52.0 (±24.0) | 500.0 (±137.2) | 1282.5 (±234.1) |
| | 3DSynthFlow | **5.5** (±6.4) | **70.5** (±41.7) | **307.0** (±158.4) | **2198.5** (±54.5) | **5145.5** (±62.9) |
| ALDH1 | RxnFlow | 4.5 (±2.1) | 26.5 (±7.8) | 73.5 (±33.2) | 472.5 (±99.7) | 1240.0 (±75.0) |
| | 3DSynthFlow | **18.5** (±14.8) | **112.0** (±94.8) | **326.5** (±316.1) | **1789.5** (±989.2) | **5701.0** (±1328.4) |
| ESR_ago | RxnFlow | **5.0** (±4.2) | 17.0 (±4.2) | 44.0 (±8.5) | 440.5 (±60.1) | 1174.0 (±100.4) |
| | 3DSynthFlow | **5.0** (±2.8) | **55.0** (±43.8) | **229.5** (±200.1) | **1742.0** (±881.1) | **5015.5** (±1680.7) |
| ESR_antago | RxnFlow | 3.0 (±2.8) | 14.5 (±0.7) | 28.0 (±0.0) | 218.0 (±21.2) | 559.0 (±50.9) |
| | 3DSynthFlow | **6.5** (±4.9) | **60.0** (±11.3) | **172.5** (±40.3) | **1020.5** (±24.7) | **2977.5** (±577.7) |
| FEN1 | RxnFlow | 3.5 (±0.7) | 29.0 (±18.4) | 75.5 (±57.3) | 372.0 (±46.7) | 1057.5 (±38.9) |
| | 3DSynthFlow | **12.5** (±9.2) | **58.5** (±29.0) | **176.5** (±95.5) | **852.0** (±63.6) | **3322.0** (±49.5) |
| Average | RxnFlow | 3.8 | 21.7 | 54.6 | 400.6 | 1062.6 |
| | 3DSynthFlow | **9.6** | **71.2** | **242.4** | **1520.5** | **4432.3** |

## F.7. Full results for LIT-PCBA target

*Table 17.* Average Vina docking score for top-100 diverse modes generated against 15 LIT-PCBA targets. The best results are in bold.

| Category | Method | Average Vina Docking Score (kcal/mol, ↓) | | | | |
|---|---|---|---|---|---|---|
| | | ADRB2 | ALDH1 | ESR_ago | ESR_antago | FEN1 |
| Fragment | FragGFN | -10.19 (± 0.33) | -10.43 (± 0.29) | -9.81 (± 0.09) | -9.85 (± 0.13) | -7.67 (± 0.71) |
| | FragGFN+SA | -9.70 (± 0.61) | -9.83 (± 0.65) | -9.27 (± 0.95) | -10.06 (± 0.30) | -7.26 (± 0.10) |
| Reaction | SynNet | -8.03 (± 0.26) | -8.81 (± 0.21) | -8.88 (± 0.13) | -8.52 (± 0.16) | -6.36 (± 0.09) |
| | BBAR | -9.95 (± 0.04) | -10.06 (± 0.14) | -9.97 (± 0.03) | -9.92 (± 0.05) | -6.84 (± 0.07) |
| | SynFlowNet[a] | -10.85 (± 0.10) | -10.69 (± 0.09) | -10.44 (± 0.05) | -10.27 (± 0.04) | -7.47 (± 0.02) |
| | SynFlowNet[b] | -9.17 (± 0.68) | -9.37 (± 0.29) | -9.17 (± 0.12) | -9.05 (± 0.14) | -6.45 (± 0.13) |
| | RGFN | -9.84 (± 0.21) | -9.93 (± 0.11) | -9.99 (± 0.11) | -9.72 (± 0.14) | -6.92 (± 0.06) |
| | RxnFlow | -11.45 (± 0.05) | -11.26 (± 0.07) | -11.15 (± 0.02) | -10.77 (± 0.04) | -7.66 (± 0.02) |
| 3D Reaction | **3DSynthFlow** | **-11.97** (± 0.12) | **-11.82** (± 0.04) | **-11.58** (± 0.07) | **-11.23** (± 0.08) | **-7.79** (± 0.01) |
| | | GBA | IDH1 | KAT2A | MAPK1 | MTORC1 |
| Fragment | FragGFN | -8.76 (± 0.46) | -9.91 (± 0.32) | -9.27 (± 0.20) | -8.93 (± 0.18) | -10.51 (± 0.31) |
| | FragGFN+SA | -8.92 (± 0.27) | -9.76 (± 0.64) | -9.14 (± 0.43) | -8.28 (± 0.40) | -10.14 (± 0.30) |
| Reaction | SynNet | -7.60 (± 0.09) | -8.74 (± 0.08) | -7.64 (± 0.38) | -7.33 (± 0.14) | -9.30 (± 0.45) |
| | BBAR | -8.70 (± 0.05) | -9.84 (± 0.09) | -8.54 (± 0.06) | -8.49 (± 0.07) | -10.07 (± 0.16) |
| | SynFlowNet[a] | -9.27 (± 0.06) | -10.40 (± 0.08) | -9.41 (± 0.04) | -8.92 (± 0.05) | -10.84 (± 0.03) |
| | SynFlowNet[b] | -8.28 (± 0.15) | -9.18 (± 0.35) | -8.06 (± 0.15) | -7.89 (± 0.13) | -9.60 (± 0.16) |
| | RGFN | -8.48 (± 0.06) | -9.49 (± 0.13) | -8.53 (± 0.11) | -8.22 (± 0.15) | -9.89 (± 0.06) |
| | RxnFlow | -9.62 (± 0.04) | -10.95 (± 0.05) | -9.73 (± 0.03) | -9.30 (± 0.01) | -11.39 (± 0.09) |
| 3D Reaction | **3DSynthFlow** | **-9.90** (± 0.14) | **-11.28** (± 0.15) | **-10.17** (± 0.37) | **-9.61** (± 0.11) | **-11.91** (± 0.01) |
| | | OPRK1 | PKM2 | PPARG | TP53 | VDR |
| Fragment | FragGFN | -10.28 (± 0.15) | -11.24 (± 0.27) | -9.54 (± 0.12) | -7.90 (± 0.02) | -10.96 (± 0.06) |
| | FragGFN+SA | -9.58 (± 0.44) | -10.83 (± 0.34) | -9.19 (± 0.29) | -7.61 (± 0.27) | -10.66 (± 0.61) |
| Reaction | SynNet | -8.70 (± 0.36) | -9.55 (± 0.14) | -7.47 (± 0.34) | -5.34 (± 0.23) | -10.98 (± 0.57) |
| | BBAR | -9.84 (± 0.10) | -11.39 (± 0.08) | -8.69 (± 0.10) | -7.05 (± 0.09) | -11.07 (± 0.04) |
| | SynFlowNet[a] | -10.34 (± 0.07) | -11.98 (± 0.12) | -9.40 (± 0.05) | -7.90 (± 0.10) | -11.62 (± 0.13) |
| | SynFlowNet[b] | -9.36 (± 0.25) | -10.64 (± 0.19) | -8.25 (± 0.10) | -6.84 (± 0.06) | -10.32 (± 0.07) |
| | RGFN | -9.61 (± 0.11) | -10.96 (± 0.18) | -8.53 (± 0.07) | -7.07 (± 0.06) | -10.86 (± 0.11) |
| | RxnFlow | -10.84 (± 0.03) | -12.53 (± 0.02) | -9.73 (± 0.02) | -8.09 (± 0.06) | -12.30 (± 0.07) |
| 3D Reaction | **3DSynthFlow** | **-11.26** (± 0.41) | **-13.36** (± 0.03) | **-10.00** (± 0.04) | **-8.41** (± 0.17) | **-12.98** (± 0.10) |

*Table 18.* Average ligand efficiency for top-100 diverse modes generated against 15 LIT-PCBA targets. The best results are in bold.

| Category | Method | Average Ligand Efficiency (↓) | | | | |
|---|---|---|---|---|---|---|
| | | ADRB2 | ALDH1 | ESR_ago | ESR_antago | FEN1 |
| Fragment | FragGFN | 0.410 (± 0.006) | 0.368 (± 0.007) | 0.347 (± 0.003) | 0.358 (± 0.002) | 0.246 (± 0.004) |
| | FragGFN+SA | 0.406 (± 0.007) | 0.374 (± 0.023) | 0.369 (± 0.003) | 0.345 (± 0.024) | 0.210 (± 0.004) |
| Reaction | SynNet | 0.274 (± 0.041) | 0.272 (± 0.006) | 0.317 (± 0.005) | 0.289 (± 0.020) | 0.196 (± 0.003) |
| | BBAR | 0.412 (± 0.006) | 0.401 (± 0.008) | 0.380 (± 0.001) | 0.387 (± 0.003) | 0.257 (± 0.003) |
| | SynFlowNet[a] | 0.401 (± 0.006) | 0.380 (± 0.007) | 0.361 (± 0.003) | 0.361 (± 0.004) | 0.247 (± 0.004) |
| | Synflownet[b] | 0.380 (± 0.021) | 0.362 (± 0.016) | 0.351 (± 0.008) | 0.349 (± 0.005) | 0.234 (± 0.007) |
| | RGFN | 0.393 (± 0.005) | 0.357 (± 0.004) | 0.346 (± 0.002) | 0.344 (± 0.002) | 0.241 (± 0.001) |
| | RxnFlow | 0.412 (± 0.005) | 0.396 (± 0.005) | 0.375 (± 0.002) | 0.380 (± 0.004) | 0.246 (± 0.001) |
| 3D Reaction | **3DSynthFlow** | **0.448** (± 0.019) | **0.395** (± 0.006) | **0.391** (± 0.005) | **0.398** (± 0.016) | **0.252** (± 0.005) |
| | | GBA | IDH1 | KAT2A | MAPK1 | MTORC1 |
| Fragment | FragGFN | 0.333 (± 0.018) | 0.367 (± 0.009) | 0.322 (± 0.008) | 0.302 (± 0.002) | 0.354 (± 0.005) |
| | FragGFN+SA | 0.318 (± 0.005) | 0.369 (± 0.020) | 0.298 (± 0.020) | 0.294 (± 0.015) | 0.355 (± 0.027) |
| Reaction | SynNet | 0.244 (± 0.013) | 0.281 (± 0.016) | 0.294 (± 0.042) | 0.226 (± 0.004) | 0.316 (± 0.035) |
| | BBAR | 0.336 (± 0.002) | 0.382 (± 0.005) | 0.332 (± 0.003) | 0.320 (± 0.002) | 0.385 (± 0.004) |
| | SynFlowNet[a] | 0.330 (± 0.004) | 0.368 (± 0.002) | 0.327 (± 0.003) | 0.305 (± 0.002) | 0.368 (± 0.002) |
| | SynFlowNet[b] | 0.324 (± 0.007) | 0.360 (± 0.013) | 0.309 (± 0.004) | 0.297 (± 0.011) | 0.361 (± 0.009) |
| | RGFN | 0.310 (± 0.002) | 0.351 (± 0.003) | 0.310 (± 0.003) | 0.298 (± 0.002) | 0.346 (± 0.004) |
| | RxnFlow | 0.327 (± 0.004) | 0.378 (± 0.001) | 0.330 (± 0.001) | 0.313 (± 0.001) | 0.370 (± 0.001) |
| 3D Reaction | **3DSynthFlow** | **0.340** (± 0.018) | **0.388** (± 0.007) | **0.340** (± 0.016) | **0.322** (± 0.009) | **0.387** (± 0.008) |
| | | OPRK1 | PKM2 | PPARG | TP53 | VDR |
| Fragment | FragGFN | 0.352 (± 0.004) | 0.442 (± 0.008) | 0.319 (± 0.007) | 0.307 (± 0.005) | 0.394 (± 0.006) |
| | FragGFN+SA | 0.327 (± 0.014) | 0.440 (± 0.009) | 0.303 (± 0.013) | 0.248 (± 0.025) | 0.390 (± 0.020) |
| Reaction | SynNet | 0.298 (± 0.039) | 0.296 (± 0.005) | 0.253 (± 0.031) | 0.211 (± 0.031) | 0.359 (± 0.015) |
| | BBAR | 0.370 (± 0.006) | 0.442 (± 0.004) | 0.326 (± 0.007) | 0.288 (± 0.005) | 0.409 (± 0.002) |
| | SynFlowNet[a] | 0.359 (± 0.004) | 0.427 (± 0.003) | 0.317 (± 0.002) | 0.287 (± 0.008) | 0.393 (± 0.003) |
| | SynFlowNet[b] | 0.355 (± 0.008) | 0.410 (± 0.018) | 0.296 (± 0.010) | 0.282 (± 0.005) | 0.380 (± 0.005) |
| | RGFN | 0.349 (± 0.001) | 0.405 (± 0.002) | 0.307 (± 0.002) | 0.271 (± 0.001) | 0.381 (± 0.002) |
| | RxnFlow | 0.369 (± 0.007) | 0.436 (± 0.005) | 0.319 (± 0.002) | 0.289 (± 0.003) | 0.405 (± 0.002) |
| 3D Reaction | **3DSynthFlow** | **0.389** (± 0.006) | **0.444** (± 0.010) | **0.321** (± 0.005) | **0.294** (± 0.006) | **0.416** (± 0.009) |

*Table 19.* Average success rate of AiZynthFinder for top-100 diverse modes generated against 15 LIT-PCBA targets. The best results are in bold.

| Category | Method | AiZynthFinder Success Rate (%, ↑) | | | | |
|---|---|---|---|---|---|---|
| | | ADRB2 | ALDH1 | ESR_ago | ESR_antago | FEN1 |
| Fragment | FragGFN | 4.00 (± 3.54) | 3.75 (± 1.92) | 1.00 (± 1.00) | 3.75 (± 1.92) | 0.25 (± 0.43) |
| | FragGFN+SA | 5.75 (± 1.48) | 6.00 (± 2.55) | 4.00 (± 2.24) | 1.00 (± 0.00) | 0.00 (± 0.00) |
| Reaction | SynNet | 54.17 (± 7.22) | 50.00 (± 0.00) | 50.00 (± 0.00) | 25.00 (± 25.00) | 50.00 (± 0.00) |
| | BBAR | 21.25 (± 5.36) | 19.50 (± 3.20) | 17.50 (± 1.50) | 19.50 (± 3.64) | 20.00 (± 2.12) |
| | SynFlowNet[a] | 52.75 (± 1.09) | 57.00 (± 6.04) | 53.75 (± 9.52) | 56.50 (± 2.29) | 53.00 (± 8.92) |
| | SynFlowNet[b] | 56.50 (± 6.58) | 56.00 (± 3.08) | 61.00 (± 2.74) | 64.50 (± 9.86) | 60.75 (± 3.77) |
| | RGFN | 46.75 (± 6.86) | 47.50 (± 4.06) | 50.25 (± 2.17) | 49.25 (± 4.38) | 48.50 (± 6.58) |
| | RxnFlow | 60.25 (± 3.77) | 63.25 (± 3.11) | 71.25 (± 4.15) | 66.50 (± 4.03) | **65.50** (± 4.09) |
| 3D Reaction | **3DSynthFlow** | **85.00** (± 7.87) | **69.33** (± 3.86) | **80.00** (± 4.24) | **69.67** (± 14.06) | 60.00 (± 15.64) |
| | | GBA | IDH1 | KAT2A | MAPK1 | MTORC1 |
| Fragment | FragGFN | 5.00 (± 4.24) | 4.50 (± 1.66) | 1.25 (± 0.83) | 0.75 (± 0.83) | 2.75 (± 1.30) |
| | FragGFN+SA | 3.00 (± 1.00) | 4.50 (± 4.97) | 1.50 (± 0.50) | 3.25 (± 1.48) | 3.50 (± 2.50) |
| Reaction | SynNet | 50.00 (± 0.00) | 50.00 (± 0.00) | 45.83 (± 27.32) | 50.00 (± 0.00) | 54.17 (± 7.22) |
| | BBAR | 17.75 (± 2.28) | 19.50 (± 1.50) | 18.75 (± 1.92) | 16.25 (± 3.49) | 18.75 (± 3.90) |
| | SynFlowNet[a] | 58.00 (± 4.64) | 59.00 (± 4.06) | 55.50 (± 10.23) | 47.25 (± 6.61) | 57.00 (± 7.58) |
| | SynFlowNet[b] | 61.50 (± 3.84) | 60.50 (± 3.91) | 57.25 (± 4.97) | 44.50 (± 9.29) | 62.00 (± 1.22) |
| | RGFN | 48.00 (± 1.22) | 43.00 (± 2.74) | 49.00 (± 1.22) | 42.00 (± 3.00) | 44.50 (± 4.03) |
| | RxnFlow | **66.00** (± 1.58) | 64.00 (± 5.05) | 66.50 (± 2.06) | **63.00** (± 4.64) | **70.50** (± 2.87) |
| 3D Reaction | **3DSynthFlow** | 64.67 (± 3.09) | **65.67** (± 13.91) | **68.00** (± 19.51) | 54.67 (± 10.50) | 64.67 (± 15.69) |
| | | OPRK1 | PKM2 | PPARG | TP53 | VDR |
| Fragment | FragGFN | 2.50 (± 2.29) | 8.75 (± 3.11) | 0.75 (± 0.43) | 4.25 (± 1.64) | 3.50 (± 2.18) |
| | FragGFN+SA | 3.25 (± 1.79) | 9.75 (± 2.28) | 1.25 (± 1.09) | 2.25 (± 1.92) | 3.75 (± 2.77) |
| Reaction | SynNet | 54.17 (± 7.22) | 50.00 (± 0.00) | 54.17 (± 7.22) | 29.17 (± 18.16) | 45.83 (± 7.22) |
| | BBAR | 13.75 (± 3.11) | 20.00 (± 0.71) | 15.50 (± 2.29) | 18.50 (± 3.28) | 12.25 (± 3.34) |
| | SynFlowNet[a] | 56.50 (± 7.63) | 50.75 (± 1.09) | 53.50 (± 5.68) | 55.50 (± 9.94) | 53.50 (± 1.80) |
| | SynFlowNet[b] | 56.25 (± 2.49) | 58.00 (± 7.00) | 57.00 (± 5.74) | 66.50 (± 6.80) | 53.50 (± 3.84) |
| | RGFN | 48.00 (± 2.55) | 48.50 (± 3.20) | 47.00 (± 5.83) | 53.25 (± 3.63) | 46.50 (± 2.69) |
| | RxnFlow | **72.25** (± 2.05) | 62.00 (± 3.24) | 65.50 (± 4.03) | **67.50** (± 2.96) | 66.75 (± 2.28) |
| 3D Reaction | **3DSynthFlow** | 67.33 (± 20.37) | **78.67** (± 8.73) | **65.67** (± 8.18) | 61.33 (± 11.73) | **74.00** (± 17.15) |

*Table 20.* Average synthesis steps estimated from AiZynthFinder for top-100 diverse modes generated against 15 LIT-PCBA targets. The best results are in bold.

| Category | Method | Average Number of Synthesis Steps ($\downarrow$) | | | | |
|---|---|---|---|---|---|---|
| | | ADRB2 | ALDH1 | ESR_ago | ESR_antago | FEN1 |
| Fragment | FragGFN | 3.60 ($\pm$ 0.10) | 3.83 ($\pm$ 0.08) | 3.76 ($\pm$ 0.20) | 3.76 ($\pm$ 0.16) | 3.34 ($\pm$ 0.18) |
| | FragGFN+SA | 3.73 ($\pm$ 0.09) | 3.66 ($\pm$ 0.04) | 3.66 ($\pm$ 0.07) | 3.67 ($\pm$ 0.21) | 3.79 ($\pm$ 0.19) |
| Reaction | SynNet | 3.29 ($\pm$ 0.36) | 3.50 ($\pm$ 0.00) | 3.00 ($\pm$ 0.00) | 4.13 ($\pm$ 0.89) | 3.50 ($\pm$ 0.00) |
| | BBAR | 3.60 ($\pm$ 0.17) | 3.62 ($\pm$ 0.19) | 3.76 ($\pm$ 0.04) | 3.72 ($\pm$ 0.11) | 3.59 ($\pm$ 0.14) |
| | SynFlowNet[a] | 2.64 ($\pm$ 0.07) | 2.48 ($\pm$ 0.07) | 2.60 ($\pm$ 0.25) | 2.45 ($\pm$ 0.09) | 2.56 ($\pm$ 0.29) |
| | SynFlowNet[b] | **2.42** ($\pm$ 0.10) | 2.48 ($\pm$ 0.10) | 2.38 ($\pm$ 0.10) | 2.34 ($\pm$ 0.30) | 2.41 ($\pm$ 0.14) |
| | RGFN | 2.88 ($\pm$ 0.21) | 2.65 ($\pm$ 0.09) | 2.78 ($\pm$ 0.19) | 2.91 ($\pm$ 0.23) | 2.76 ($\pm$ 0.17) |
| | RxnFlow | 2.42 ($\pm$ 0.23) | **2.19** ($\pm$ 0.12) | **1.95** ($\pm$ 0.20) | **2.15** ($\pm$ 0.18) | **2.23** ($\pm$ 0.18) |
| 3D Reaction | **3DSynthFlow** | **1.77** ($\pm$ 0.37) | 2.23 ($\pm$ 0.16) | 2.02 ($\pm$ 0.20) | 2.43 ($\pm$ 0.61) | 2.73 ($\pm$ 0.50) |
| | | GBA | IDH1 | KAT2A | MAPK1 | MTORC1 |
| Fragment | FragGFN | 3.94 ($\pm$ 0.11) | 3.74 ($\pm$ 0.10) | 3.78 ($\pm$ 0.09) | 3.72 ($\pm$ 0.18) | 3.84 ($\pm$ 0.18) |
| | FragGFN+SA | 3.94 ($\pm$ 0.15) | 3.84 ($\pm$ 0.23) | 3.66 ($\pm$ 0.18) | 3.69 ($\pm$ 0.21) | 3.94 ($\pm$ 0.08) |
| Reaction | SynNet | 3.38 ($\pm$ 0.22) | 3.38 ($\pm$ 0.22) | 3.46 ($\pm$ 0.95) | 3.50 ($\pm$ 0.00) | 3.29 ($\pm$ 0.36) |
| | BBAR | 3.71 ($\pm$ 0.12) | 3.68 ($\pm$ 0.02) | 3.63 ($\pm$ 0.05) | 3.73 ($\pm$ 0.05) | 3.77 ($\pm$ 0.09) |
| | SynFlowNet[a] | 2.48 ($\pm$ 0.18) | 2.61 ($\pm$ 0.13) | 2.45 ($\pm$ 0.37) | 2.81 ($\pm$ 0.24) | 2.44 ($\pm$ 0.27) |
| | SynFlowNet[b] | 2.45 ($\pm$ 0.08) | 2.46 ($\pm$ 0.12) | 2.45 ($\pm$ 0.12) | 2.83 ($\pm$ 0.27) | 2.39 ($\pm$ 0.17) |
| | RGFN | 2.77 ($\pm$ 0.20) | 2.97 ($\pm$ 0.15) | 2.78 ($\pm$ 0.10) | 2.86 ($\pm$ 0.19) | 2.92 ($\pm$ 0.06) |
| | RxnFlow | **2.10** ($\pm$ 0.08) | **2.16** ($\pm$ 0.11) | **2.29** ($\pm$ 0.05) | **2.29** ($\pm$ 0.11) | **2.05** ($\pm$ 0.09) |
| 3D Reaction | **3DSynthFlow** | 2.47 ($\pm$ 0.24) | 2.49 ($\pm$ 0.42) | 2.47 ($\pm$ 0.61) | 2.70 ($\pm$ 0.33) | 2.42 ($\pm$ 0.61) |
| | | OPRK1 | PKM2 | PPARG | TP53 | VDR |
| Fragment | FragGFN | 3.82 ($\pm$ 0.13) | 3.71 ($\pm$ 0.12) | 3.73 ($\pm$ 0.24) | 3.73 ($\pm$ 0.23) | 3.75 ($\pm$ 0.06) |
| | FragGFN+SA | 3.62 ($\pm$ 0.12) | 3.84 ($\pm$ 0.21) | 3.71 ($\pm$ 0.04) | 3.66 ($\pm$ 0.05) | 3.67 ($\pm$ 0.25) |
| Reaction | SynNet | 3.29 ($\pm$ 0.36) | 3.50 ($\pm$ 0.00) | 3.29 ($\pm$ 0.36) | 3.67 ($\pm$ 0.91) | 3.63 ($\pm$ 0.22) |
| | BBAR | 3.70 ($\pm$ 0.17) | 3.61 ($\pm$ 0.05) | 3.72 ($\pm$ 0.13) | 3.65 ($\pm$ 0.05) | 3.77 ($\pm$ 0.16) |
| | SynFlowNet[a] | 2.49 ($\pm$ 0.33) | 2.62 ($\pm$ 0.10) | 2.56 ($\pm$ 0.12) | 2.51 ($\pm$ 0.27) | 2.55 ($\pm$ 0.09) |
| | SynFlowNet[b] | 2.50 ($\pm$ 0.11) | 2.52 ($\pm$ 0.20) | 2.53 ($\pm$ 0.06) | 2.34 ($\pm$ 0.10) | 2.51 ($\pm$ 0.17) |
| | RGFN | 2.81 ($\pm$ 0.12) | 2.82 ($\pm$ 0.10) | 2.82 ($\pm$ 0.18) | 2.64 ($\pm$ 0.10) | 2.84 ($\pm$ 0.18) |
| | RxnFlow | **2.00** ($\pm$ 0.09) | 2.34 ($\pm$ 0.19) | **2.21** ($\pm$ 0.06) | **2.12** ($\pm$ 0.12) | **2.12** ($\pm$ 0.12) |
| 3D Reaction | **3DSynthFlow** | 2.34 ($\pm$ 0.53) | **2.20** ($\pm$ 0.41) | 2.61 ($\pm$ 0.39) | 2.73 ($\pm$ 0.47) | 2.25 ($\pm$ 0.78) |

## F.8. Example generation trajectories

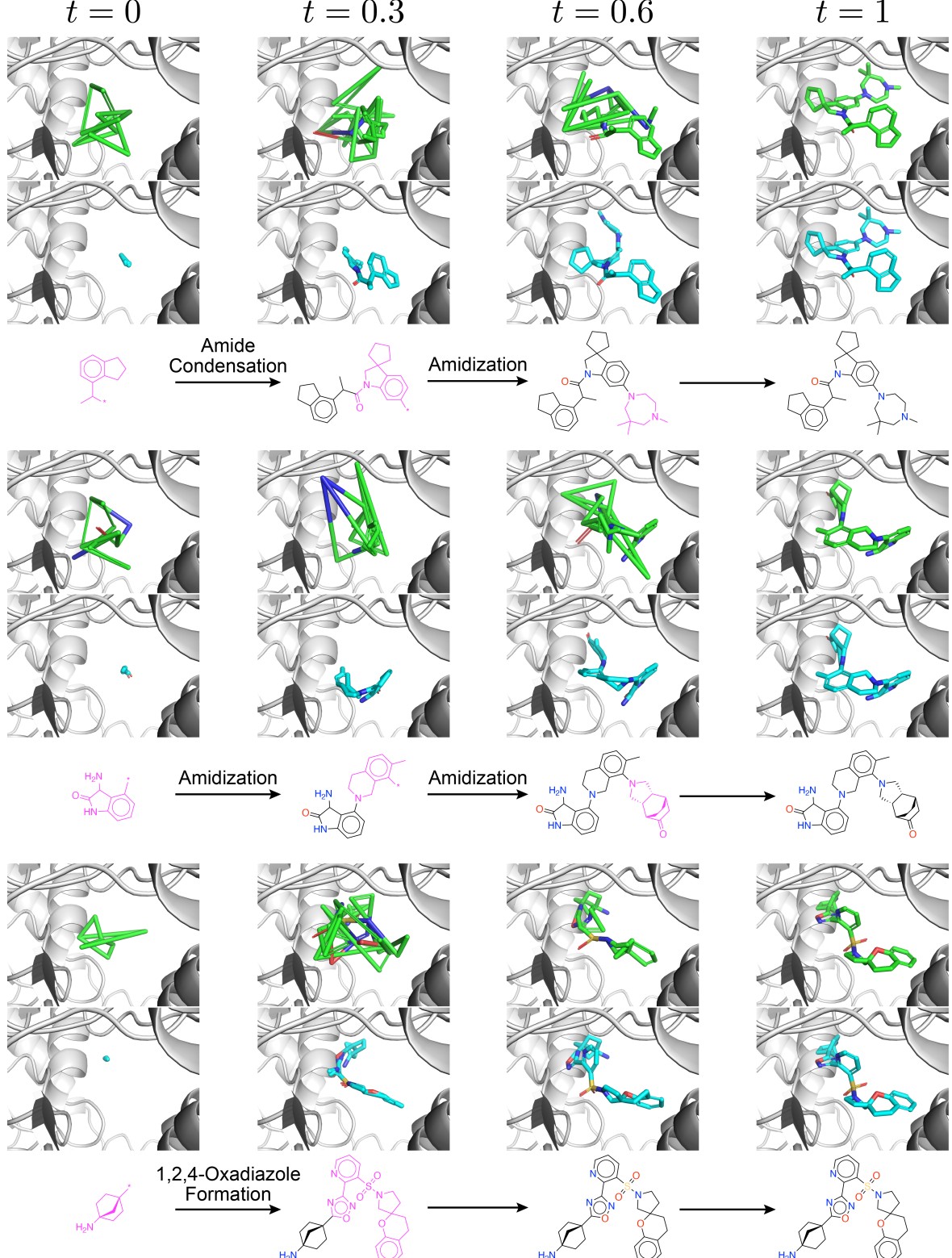

*Figure 7.* The example generation trajectory against ALDH1 target. The top row shows the 3D molecule $x_t$ (green). The mid row shows the predicted final pose at each time step (cyan). The bottom row shows the synthesis pathway.

