# OpenReview forum: "Compositional Flows for 3D Molecule and Synthesis Pathway Co-design"
_ICML.cc/2025/Conference — ICML 2025 poster_

### Official Review · Reviewer_ScwC · 2025-03-11

**Overall Recommendation:** 3

**Summary:**

The paper introduces a novel flow matching framework, 3DSynthFlow, for generating synthesizable molecules within protein pockets by sequentially selecting discrete building blocks and simultaneously modeling their coordinates. The authors evaluate 3DSynthFlow against all 15 protein targets in the LIT-PCBA virtual screening benchmark.

**Claims And Evidence:**

The claims are adequately supported by the evidence presented in the paper.

**Essential References Not Discussed:**

N/A

**Experimental Designs Or Analyses:**

Discussed below.

**Methods And Evaluation Criteria:**

The authors evaluate 3DSynthFlow on targets in the LIT-PCBA dataset using docking as an oracle. The exhaustive validation of the model on several protein targets and its consistent strong performance strengthens the results of the paper. However, the evaluation of the generated molecules for diversity and respective properties is somewhat lacking, discussed below in the Weaknesses section.

**Other Comments Or Suggestions:**

N/A

**Other Strengths And Weaknesses:**

Strengths:
- This reviewer notes that the 3D information provided by the state flow model allows for better-informed building block selections with the GFlowNet at intermediate steps during molecule generation compared to 2D methods.
- This reviewer agrees that reaction template-based generation constraints are important for practical molecular generation in 3D, and finds the sequential design apt for ensuring synthesizability while retaining the ability to denoise building block coordinates according to their respective local time steps.

Weaknesses:
- The evaluation of the diversity and chemical properties of generated molecules is insufficient. The authors do not report the diversity of molecules of the 3D co-design model (as measured by average Tanimoto similarity or number of high-scoring modes, etc.) compared to the 2D models or compare these metrics between molecules generated for different protein targets.
- The approach seems to require training two separate models, the GFlowNet and the state flow (flow matching) model, which may prove somewhat computationally expensive.
- The approach uses synthons rather than template-based molecular synthesis, which aids in the simplicity of combining building blocks. However, they restrict their synthesis process to a brick-and-linker formulation with up to 3 blocks, which limits the explored chemical space to linear molecules.

**Questions For Authors:**

- This reviewer is curious about the exploration space of the enamine synthons and requests that the authors provide an estimation of the state space size similar to that of Fig. 2 of SynFlowNet or Fig. 2 of RGFN. Additionally, the reviewer would like to see the number of unique building blocks explored for experiments, similar to Fig. 7 of SynFlowNet.
- This reviewer would like more clarification as to the training details of the state flow model. In the appendix, there is mention of "decomposing CrossDocked molecules" to train the state flow model. However, it is unclear whether the training dataset for the state flow model contains "partial" CrossDocked molecules, which would aid in learning to dock the initial, individual fragments selected by the GFlowNet. This reviewer is also curious as to the success rate in decomposing molecules from CrossDocked into Enamine synthons, and what proportion of molecules succeed or fail to be decomposed.

**Relation To Broader Scientific Literature:**

The paper builds on previous literature on synthesizable molecule generation with GFlowNets, and represents a step forward in template-based molecular synthesis with a new 3-dimensional flow matching component for atomic coordinates of selected building blocks.

**Theoretical Claims:**

N/A

---

> ### Author Rebuttal · Authors · 2025-04-01
>
> We appreciate the reviewer for volunteering their valuable time and providing insightful feedback to our paper. We are addressing their questions one by one in our response below.
>
> > W1. Evaluation of the diversity and chemical properties.
>
> To further address the reviewer’s suggestion, we now compare sampling efficiency, diversity and other chemical properties of our method (CGFlow) against the 2D baseline (RxnFlow) on the first 5 LIT-PCBA targets. We define high-scoring modes as those with QED > 0.5, Vina < -10 kcal/mol for all pockets except FEN1, which we use -7 kcal/mol, and mode similarity < 0.5.
>
> The table below shows the number of unique high-scoring modes identified after sampling 10k molecules:
>
> | |ADRB2|ALDH1|ESR_ago|ESR_antago|FEN1|Avg|
> | - | - | - | - | - | - | - |
> |RxnFlow|69|97|38|28|116|69.6|
> |CGFlow|276|417|358|213|358|324.4|
>
> CGFlow consistently outperforms RxnFlow in sampling efficiency, discovering 4.7x more diverse modes. Since mode diversity increases experimental success, this improvement highlights the practical advantage of our approach.
>
> We further report the full sampling trend for ADRB2 and the average properties of the top 100 diverse modes. Diversity here is computed without similarity-based filtering to avoid artificial inflation. Our results confirm CGFlow's superior efficiency in discovering diverse modes with good QED and Vina scores.
>
> |\# of mol explored|1000|10000|64000|Vina|QED|MW|HAC|LogP|Diversity|
> |-|-|-|-|-|-|-|-|-|-|
> |RxnFlow|2|69|1448| -11.57|0.67|388.5|28.2|4.25|0.88|
> |CGFlow|20|276|4323| -12.34|0.69|386.1|27.9|4.38|0.85|
>
> We thank the reviewer for their comment, which motivated us to further highlight the strengths of our approach.
>
> > W2. Concern about computational cost.
>
> We kindly refer the reviewer to our response to Reviewer Z8PT (Weakness 2) for further clarification on computational cost.
>
> > W3. Concern about limited search space due to the brick-and-linker formulation.
>
> We appreciate the reviewer highlighting this important limitation regarding our synthesis approach. Indeed, as the reviewer pointed out, our current brick-and-linker formulation restricts the explored chemical space to linear molecules by excluding nonlinear reactions, such as ring formation. However, this constraint can be substantially mitigated by expanding the chemical search space through incorporating a larger building block library. For example, V-SYNTHES [1], a pioneering work in exploring Enamine REAL Space using a brick-and-linker strategy, successfully achieved a notable hit rate of 33%.
>
> > Q1. Question about the size of state space and unique building blocks explored during training.
>
> We estimate the sample space according to the number of synthetic steps: $10^{11}$ molecules with a single reaction step, $10^{17}$ molecules with two reaction steps, and $10^{23}$ molecules with three reaction steps. In our experiments, we employed up to two reaction steps according to Enamine REAL, and the state space size is similar to RGFN (up to 4 steps with 8,350 blocks) and SynFlowNet (up to 3 steps with 200k blocks).
>
> Additionally, we analyzed the number of unique building blocks (BBs) explored during training across the first 5 LIT-PCBA targets. Our model explored an average of ~55,000 unique BBs within 1,000 training iterations with a batch size of 64. This demonstrates a significantly broader exploration compared to SynFlowNet, which reported exploring ~15,000 unique BBs during 8,000 training iterations with a batch size of 8.
>
> |Target|ADRB2|ALDH1|ESR_ago|ESR_antago|FEN1|
> |-|-|-|-|-|-|
> |Number of Unique blocks|$45520\pm7876$|$48644\pm1983$|$55211\pm5611$|$58097\pm8529$|$69400\pm5259$|
>
> > Q2. clarification as to the training details of the state flow model.
>
> Yes, you are absolutely correct that during training, the state flow model encounters "partial" CrossDocked molecules. Specifically, we decompose each molecule into up to three fragments using 38 bimolecular Enamine synthesis protocols defined by reaction SMARTS. We then randomly sample a fragment ordering, matching the fragment introduction schedule used in the Compositional Flow model (e.g., fragment A at $t=0$, B at $t=0.3$, etc.). A random time step is sampled so that at earlier steps, the model sees "partial" structures. This design enables the state flow model to learn realistic fragment docking conformations aligned with the fragment selection process of compositional flow.
>
> Importantly, the decomposition is not intended to recover purchasable Enamine synthons but to expose the model to chemically meaningful substructures for learning protein-pocket conformations. While guided by Enamine protocols, exact synthon matching is unnecessary, allowing us to use any molecules for pose prediction training. We will clarify these points in the revised appendix.
>
> Reference:
> 1. Sadybekov, Arman A., et al. "Synthon-based ligand discovery in virtual libraries of over 11 billion compounds." Nature 601.7893 (2022): 452-459.

---

### Official Review · Reviewer_eDTU · 2025-03-14

**Overall Recommendation:** 1

**Summary:**

The paper introduces Compositional Generative Flows (CGFlow), a novel framework designed for the generation of compositional objects with continuous features in generative applications, such as synthesis-based 3D molecular design. CGFlow extends flow matching by enabling the generation of objects in compositional steps while modeling continuous states. This is accomplished through a straightforward expansion of the flow matching interpolation process to model compositional state transitions. Additionally, CGFlow builds upon the theoretical foundations of generative flow networks (GFlowNets), allowing for reward-guided sampling of compositional structures.

The framework is applied to synthesizable drug design by simultaneously designing both the molecule's synthetic pathway and its 3D binding pose. CGFlow achieves state-of-the-art binding affinity compared to synthesis-based baselines, demonstrated across all 15 targets of the LIT-PCBA benchmark. Further evaluation with PoseCheck indicates that molecules designed using CGFlow exhibit a higher number of key protein-ligand interactions, underscoring the benefits of co-designing 3D molecular structures alongside their synthetic pathways.

## update after rebuttal

I keep my original rating. The authors have provided more evaluations in their rebuttal. My main concern is that the improvement of the proposed method over RxnFlow is too marginal while it introduces much more latency.

**Claims And Evidence:**

Yes.

**Essential References Not Discussed:**

See above.

For a more comprehensive evaluation, the authors could consider comparing their CGFlow framework against structure-based drug design (SBDD) baselines, including references [1,2,3] among others. This would provide a more thorough assessment of CGFlow's performance and robustness in the context of established SBDD methodologies, enabling a clearer understanding of its strengths and potential areas of improvement.

[1] Zhang, Z. and Liu, Q., 2023, July. Learning subpocket prototypes for generalizable structure-based drug design. In International Conference on Machine Learning (pp. 41382-41398). PMLR.

[2] Zhou, X., Cheng, X., Yang, Y., Bao, Y., Wang, L. and Gu, Q., 2024. Decompopt: Controllable and decomposed diffusion models for structure-based molecular optimization. arXiv preprint arXiv:2403.13829.

[3] Qu, Y., Qiu, K., Song, Y., Gong, J., Han, J., Zheng, M., Zhou, H. and Ma, W.Y., 2024. Molcraft: Structure-based drug design in continuous parameter space. arXiv preprint arXiv:2404.12141.

**Experimental Designs Or Analyses:**

Yes. I also recommend to conduct experiments on CrossDocked2020 or BindingMOAD.

**Methods And Evaluation Criteria:**

Yes.

**Other Comments Or Suggestions:**

N/A

**Other Strengths And Weaknesses:**

Strengths:
1. This work introduces 3D information of ligands into RxnFlow-like frameworks, which enable modeling protein-ligand interactions.

Weaknesses:
1. Lack of ablation studies: since the time scheduler plays an important role in the proposed compositional flow matching, the related ablation studies are required. Currently, no related ablation studies are provided. What if the denoising process of coordinates have no overlap across different synthons?

2. Lack of comprehensive evaluation: test on more datasets, such as CrossDocked2020 and BindingMOAD; measurement of training efficiency; lack of evaluation in geometrical properties since this paper is related to 3D molecule generation.

3. Lack of baselines: SBDD baselines.

4. The performance is not good: The vina improvement over RxnFlow is marginal, but the degeneration in Success Rate (synthesizability) and synthesize steps is obvious. It is well-known that the ligands with larger molecular weights tend to have better Vina scores. So it is suspectable that the slight improvement in Vina score comes from more synthesis steps.

**Questions For Authors:**

I noticed that the evaluation results in local optimized posed and rocked. Have you evaluated the Vina score directly on the generated poses? Have you checked the RMSD between the generated poses and redocked poses.

**Relation To Broader Scientific Literature:**

The key contributions of the paper mainly relate to RxnFlow and Diffusion forcing. The work combines the ideas from both. The related works have been discussed in the paper. Additionally, I recommend the authors to cite and compare with other references which resemble this work somewhat [1].

[1] Ghorbani, Mahdi, et al. "Autoregressive fragment-based diffusion for pocket-aware ligand design." arXiv preprint arXiv:2401.05370 (2023).

**Theoretical Claims:**

I have checked all theoretical claims and found that they are correct.

---

> ### Author Rebuttal · Authors · 2025-04-01
>
> We highly appreciate this reviewer’s constructive feedback and insightful suggestions. We would like to clarify and address all of these points to the best of our ability in the response below.
>
> > W1: Lack of ablation studies about time scheduling of state flow model
>
> Thank you for the valuable suggestion. We conducted an ablation study to assess the effect of time scheduling in state flow training, comparing three settings: partial (partial overlap of synthon denoising), no overlap (strictly autoregressive), and till end (all synthons denoised until $t=1$).
>
> We compared the average local-optimized Vina docking scores across different training iterations for the ALDH1 target below:
>
> | \# of mol explored | 10,000 | 20,000 | 30,000 |
> |-|-|-|-|
> | no overlap | $-5.68 \pm 0.29$ | $-6.33 \pm 0.26$ | $-7.02 \pm 0.34$ |
> | partial | $-6.28 \pm 0.22$ | $-7.28 \pm 0.21$ | $-7.22 \pm 0.12$ |
> | till end | $-7.15 \pm 0.40$ | $-7.60 \pm 0.29$ |$ -7.79 \pm 0.12$ |
>
> CGFlow’s overlapping noise scheduling, where positions are refined as synthons are added, clearly outperforms conventional autoregressive approaches (no overlap).
>
> > W2 & W3: Lack of comprehensive evaluation (e.g., CrossDocked2020) and SBDD baselines.
>
> Following your suggestions, we evaluated CGFlow on CrossDocked2020 against established SBDD baselines. Using the same conditional objective and proxy setup as TacoGFN and RxnFlow, we generated 100 molecules per pocket in a zero-shot manner without an additional optimizing process for test targets. We varied the reward exponentiation parameter β (Low: U(1,64), Medium: U(32,64), High: U(48,64)) to balance exploitation and exploration for sampling.
>
> |   |Validity(↑)|Vina(↓)|QED(↑)|AiZyn. Succ Rate(↑)|Div(↑)|Time(↓)|
> |---|---|---|---|---|---|---|
> |Reference| - | -7.71|0.48|36.1| - | - |
> |FLAG|99.7|-7.07|0.49|21.9|0.82|1047|
> |DecompDiff|66.0|-8.35|0.37|0.9|0.84|6189|
> |MolCRAFT|96.7|-8.05|0.50|16.5|0.84|141|
> |MolCRAFT-large|70.8|-9.25|0.45|3.9|0.82 | >141|
> |TacoGFN|100.0|-8.24|0.67|1.3|0.67|4|
> |RxnFlow|100.0|-8.85|0.67|34.8|0.81|4|
> |CGFlow (low β)|100.0|-9.00|0.72|55.0|0.79|24|
> |CGFlow (med β)|100.0|-9.16|0.73|56.6|0.76|24|
> |CGFlow (high β)|100.0|-9.38|0.74|62.2|0.66|24|
>
> CGFlow reduces Vina from -8.85 (RxnFlow) to -9.38 (CGFlow-high beta), outperforming all baselines. It also yields the highest QED scores (0.72–0.74) and highest AiZynthFinder success rate (62.2%) compared to all baselines, underscoring the practical benefits of synthesis-aware generation. CGFlow shows consistent synthesis success rate across both CrossDock (55.0%–62.2%) and LIT-PCBA (53.1%) benchmarks.
>
> > W4. Concerns about reward hacking by generating larger molecules.
>
> To address this concern, we conducted additional experiments on the first five targets, restricting heavy atom count (HAC) to 40. CGFlow still outperforms RxnFlow in Vina score (-10.94 vs -10.46) with comparable HAC (29.63 vs 29.37). Moreover, CGFlow achieves the highest ligand efficiency (0.375) - computed by Vina / HAC, confirming that our binding affinity gains stem from the 3D co-design strategy rather than molecule size.
>
> | |Vina (↓)|Ligand Efficiency (↑)|Avg Heavy atom count|
> |-|-|-|-|
> |SynFlownet|-8.644|0.335|26.44|
> |RGFN|-9.085|0.329|28.02|
> |RxnFlow|-10.457|0.362|29.37|
> |CGFlow (rebuttal)|-10.940|0.375|29.63|
>
> > W2 / W4: Measurement of training efficiency / The vina improvement over RxnFlow is marginal
>
> We kindly refer the reviewer to our response to Reviewer ScwC (Weakness 1) for experimental results on training efficiency - where we show CGFlow discovers 4.7× more diverse modes than RxnFlow. We note that optimization of docking scores is restricted by the saturation of the pocket's binding interactions. At that point, discovering more diverse binding modes becomes more important to maximize the success rate of practical applications.
>
> > W4. Degeneration in Success Rate (synthesizability) and synthesize steps
>
> The small drop in AiZynthFinder synthesizability arises from our transition from reaction-based generation (RxnFlow) to a synthon-based (brick-and-linker) approach. Reaction-based generation often halts prematurely if a state molecule lacks any reactive functional groups, while the synthon-based method can easily construct molecules with longer synthetic trajectories. We emphasize that our approach use the building block and synthesis reactions from Enamine REAL and xREAL, known for a wet-lab synthetic success rate of 80%.
>
> > W2/Q1. Lack of evaluation in geometrical properties and questions regarding generated poses.
>
> We evaluated various geometrical properties and Vina score of the generated poses of the top 100 molecules of local-optimized Vina optimization across 3 seeds.
>
> |Metric|Validity|Med. Energy|Med. Strain Energy|Score|Minimize|Dock|Redock RMSD<1Å|Redock RMSD<2Å|
> |-|-|-|-|-|-|-|-|-|
> |Value|$1.0\pm0.0$|$226.89\pm18.83$|$147.82\pm17.95$|$-8.749\pm0.438$|$-12.240\pm0.127$|$-12.915\pm0.104$|$13.3 \pm2.1$%|$57.3\pm10.6$%|

---

### Official Review · Reviewer_Z8PT · 2025-03-14

**Overall Recommendation:** 5

**Summary:**

This paper introduces Compositional Generative Flows (CGFlow), a framework that extends flow matching to generate objects with compositional structures and continuous features simultaneously. CGFlow combines two interleaved processes: Compositional Flow for modeling the probability path of compositional structures and State Flow for managing continuous states associated with these structures. The authors apply CGFlow to drug design through 3DSynthFlow, which jointly designs molecules' 3D binding poses and their synthetic pathways. By co-designing the 3D molecular structure and synthesis pathway, 3DSynthFlow achieves state-of-the-art binding affinity across all 15 targets in the LIT-PCBA benchmark compared to synthesis-based baselines. Evaluation using PoseCheck shows that molecules designed by 3DSynthFlow exhibit more protein-ligand interactions, demonstrating the value of 3D structure and synthesis pathway co-design. This approach addresses key challenges in drug discovery by ensuring both strong binding affinity and synthesizability.

**Claims And Evidence:**

The claims in the paper are generally well-supported by clear and convincing evidence

The evidence is presented in a transparent manner with multiple evaluation metrics, statistical significance indicated by standard deviations across multiple runs, and comprehensive comparisons against relevant baseline methods.

**Essential References Not Discussed:**

not to my knowledge.

**Experimental Designs Or Analyses:**

The use of the LIT-PCBA benchmark with 15 diverse protein targets is appropriate and comprehensive.

The authors compare against a broad range of relevant baselines including fragment-based methods and reaction-based methods. This is thorough and appropriate.

However, the paper lacks details about computational requirements and training time comparisons, which is important for assessing practical applicability.

**Methods And Evaluation Criteria:**

The CGFlow framework logically extends flow matching to handle compositional objects with continuous features, which is precisely what's needed for molecular design where both structure and 3D conformation matter. The theoretical foundation connecting compositional flow with GFlowNets for discrete structure generation and state flow for continuous features is sound.

For evaluation, their choice of the LIT-PCBA benchmark is appropriate as it's a standard dataset for structure-based drug design.

The authors also conduct ablation studies on reward computation approaches and compare against multiple baselines including both fragment-based and reaction-based methods. Their experimental setup effectively demonstrates the value of jointly modeling 3D structure and synthesis pathways, which directly addresses limitations in prior work that focused on only one aspect or the other.

However, the authors claim their approach handles "compositional objects with continuous features" broadly, but their evaluation is restricted only to molecular design. This raises questions about the true generalizability of CGFlow to other domains.

**Other Comments Or Suggestions:**

NA

**Other Strengths And Weaknesses:**

- Strengths:
* The work addresses a significant real-world challenge in drug discovery by simultaneously optimizing for binding affinity and synthesizability. The practical utility is clear and valuable.
* The figures (especially Figure 1) effectively illustrate the concept of interleaving compositional structure and continuous state generation, helping readers understand a complex methodology.

- Weaknesses (just because I have to):
* The paper would be stronger with more analysis of cases where the method performs poorly or limitations in certain chemical spaces.
* The method introduces additional complexity compared to existing approaches. A more explicit discussion of the implementation challenges and computational overhead would provide a more balanced assessment of practical applicability.

Overall, the paper presents a significant contribution by addressing the important challenge of jointly modeling 3D structure and synthesizability in molecular design, with a methodologically sound approach that could inspire future work in compositional generative modeling.

**Questions For Authors:**

Could you explain the rationale behind the footnote of page 14?

**Relation To Broader Scientific Literature:**

- The authors build upon prior flow matching work by Lipman et al. (2023) and extend it to handle compositional structures
- The paper incorporates the GFlowNet framework introduced by Bengio et al. (2021) for exploring discrete compositional spaces.
- The authors connect their work to recent advances in sequential diffusion models.
- The authors address the synthesizability challenge highlighted by Gao & Coley (2020)

**Theoretical Claims:**

The main theoretical component appears in Appendix B (pages 14-15) where the authors develop the trajectory balance objective for their compositional flow model. The proof relies heavily on determinism introduced by fixing the random seed, which feels like a theoretical workaround rather than a principled approach. Please correct me if I'm wrong.

---

> ### Author Rebuttal · Authors · 2025-04-01
>
> We sincerely thank the reviewer for their thoughtful and detailed evaluation, and for recognizing the significance of our framework contribution and its application to 3D molecule and synthesis pathway co-design.
>
> > W1. More analysis of cases where the method performs poorly or limitations in certain chemical spaces.
>
> Our synthesis-based action space based on brick & linker synthons does not yet explore certain synthetic pathways such as ring-forming reactions, and nonlinear synthetic pathways, as pointed out by Reviewer ScwC. Our method predicts the 3D binding poses of intermediate states. This synthon-based generation is chosen to prevent the atoms from intermediate states being lost from reactions or forming new rings - which would degrade the accuracy of pose prediction for intermediate states.
>
> Moreover, our pose prediction module is trained on the CrossDocked2020 dataset, in which binding pockets are extracted with a distance cutoff based on the reference ligand structure, resulting in an inherently biased pocket to its reference ligand. This can negatively impact pose prediction accuracy if generated molecules optimally bind to a different subpocket from the reference ligand or are larger than the reference ligand. This issue arises during reinforcement learning-driven exploration of diverse chemical spaces. Addressing this limitation would require employing an unbiased pocket structure (e.g., using a center-based cutoff) or a full-protein structure, and we highlight this as a consideration for real-world application of our method.
>
> Finally, our training of poses is currently biased towards Vina docking poses from the CrossDocked2020 dataset. A promising direction to remove this bias is training the model on an experimental structure dataset.
>
> > W2. A more explicit discussion of the implementation challenges and computational overhead.
>
> Thank you for this suggestion! We will incorporate the following details in our final manuscript. The state-flow model (i.e., the pocket-conditional pose predictor) was trained for 100 epochs using a batch size of 32 on 4 L40 GPUs (48GB), taking a total of 18.4 hours. Importantly, as the state-flow model is trained on the CrossDocked dataset and can be reused across different test pockets, it incurs only a one-time computational cost. We also plan to release the model weights, so users will only need to train the composition-flow model tailored to their custom reward function and target.
>
> The composition-flow model is trained individually for each pocket for 1,000 steps with batch size of 64 and 80 flow matching steps. The training with GPU-accelerated docking takes 12-20 hours (depending on targets) on a single A4000 GPU (16GB). We find this computational requirement accessible for most practical drug discovery campaigns. Furthermore, the composition-flow model can also be trained in a pocket-conditioned manner (Please see SBDD benchmark in our response to Reviewer eDTU) to sample molecules for any target pocket in a zero-shot manner, making model training a one-time cost in this setting.
> |Flow matching steps|Avg Vina (↓)|Top 100 Vina (↓)|Training time (sec/iter)|Sampling time (sec/mol)|
> |-|-|-|-|-|
> |10|$-10.28\pm0.32$|$-14.27\pm0.59$|33|0.053|
> |20|$-10.24\pm0.18$|$-14.38\pm0.22$|34|0.080|
> |40|$-10.40\pm0.13$|$-14.51\pm0.27$|39|0.123|
> |60|$-10.50\pm0.14$|$-14.57\pm0.24$|44|0.160|
> |80|$-10.44\pm0.18$|$-14.53\pm0.16$|49|0.199|
>
> The sampling time of our model is 0.05~0.20 seconds/molecule depending on the choice of number of flow matching steps. We further analyzed how the number of flow matching steps impacts performance using the ALDH1 target with the Vina reward. Performance slightly improves with increased flow matching steps and saturates around 40-60 steps. We attribute this marginal improvement to the fact that the pose prediction module’s primary role is providing a spatial context between intermediate molecules and the pocket; thus, extremely precise pose predictions have limited additional impact on model decisions.
>
> > Theoretical Claim & Question
>
> We thank the reviewer for this insightful observation. Indeed, our current theoretical analysis leverages determinism through fixing the random seed primarily for analytical clarity. However, this leads to deterministic noise sampling for the initial synthon states; for example, when `C(=O)[*]` and `[*]NC` are sequentially added, their initial coordinates are identical. To maintain determinism while preventing synthons from receiving the same initial states, we can use the molecule size or a hash of the molecule to set distinct random seeds across the generation process.
>
> We acknowledge this limitation explicitly and highlight extending our theoretical framework to handle fully stochastic environments, using Expected Flow Networks [1] as a promising direction for future research.
>
> Reference:
> 1. Jiralerspong, Marco, et al. "Expected flow networks in stochastic environments and two-player zero-sum games." ICLR 2024.

---

### Decision · Program_Chairs · 2025-05-01

**Decision:**

Accept (poster)

**Comment:**

The paper on Compositional Generative Flows presents a novel and methodologically sound approach for jointly modeling 3D structure and synthesizability in molecular design, addressing a significant real-world challenge in drug discovery. Reviewers identified various strengths, such as well-supported claims, a logical extension of flow matching, and appropriate benchmarks, but also raised concerns like limited generalizability, theoretical issues, lack of certain evaluations, and performance compared to baselines. The authors made substantial efforts in their rebuttals to address these concerns, providing more details on limitations, computational aspects, conducting additional experiments, and clarifying training details.  However, after discussion, reviewers still have concerns about the effectiveness of the proposed method from the empirical perspective.
"The evaluation on CrossDock strengthens the results. However, the Vina scores of RxnFlow and CGFlow are quite similar (-8.85 versus -9.00), with RxnFlow showing a significantly faster inference time (4 versus 24). While the binding affinities are alike, CGFlow's slower generation is noteworthy. Moreover, increasing beta improves binding affinity but severely impacts diversity." After careful proof, the are chair regards this as valid, and authors should clarify this in the next version to make your manuscript stronger.